# Fast and Robust: Task Sampling with Posterior and Diversity Synergies for Adaptive Decision-Makers in Randomized Environments

**Yun Qu** [* 1]  **Qi Wang** [* 1]  **Yixiu Mao** [* 1]  **Yiqin Lv** [1]  **Xiangyang Ji** [1]

## Abstract

Task robust adaptation is a long-standing pursuit in sequential decision-making. Some risk-averse strategies, e.g., the conditional value-at-risk principle, are incorporated in domain randomization or meta reinforcement learning to prioritize difficult tasks in optimization, which demand costly intensive evaluations. The efficiency issue prompts the development of robust active task sampling to train adaptive policies, where risk-predictive models are used to surrogate policy evaluation. This work characterizes the optimization pipeline of robust active task sampling as a Markov decision process, posits theoretical and practical insights, and constitutes robustness concepts in risk-averse scenarios. Importantly, we propose an easy-to-implement method, referred to as Posterior and Diversity Synergized Task Sampling (PDTS), to accommodate fast and robust sequential decision-making. Extensive experiments show that PDTS unlocks the potential of robust active task sampling, significantly improves the zero-shot and few-shot adaptation robustness in challenging tasks, and even accelerates the learning process under certain scenarios. Our project website is at https://thu-rllab.github.io/PDTS_project_page.

## 1. Introduction

Deep reinforcement learning (RL) has garnered remarkable progress in solving complicated sequential decision-making problems in the past few years (Sutton & Barto, 2018). However, an existing challenge is effectively transferring the RL policy to unseen but similar scenarios without learning from scratch. A commonly used strategy is to randomize the environment, e.g., placing a distribution over Markov decision processes (MDPs), for policy search in a zero-shot or few-shot manner. This facilitates the rise of domain randomization (DR) (Muratore et al., 2018) and Meta-RL (Finn et al., 2017) paradigms, which train adaptive policies in task episodic learning. Simultaneously, adaptation robustness to worst-case scenarios is catching increasing attention as most real-world decision-making scenarios are inherently risk-sensitive, where failures in adaptation can cause catastrophic outcomes, e.g., damage to robots (Carpin et al., 2016) or accidents in autonomous driving (Rempe et al., 2022).

**Active Inference's Promise for Adaptive Robust Decision-Maker:** When risk-averse principles are incorporated in DR and Meta-RL to enhance adaptation robustness (Wang et al., 2024c; Lv et al., 2024; Greenberg et al., 2024); prioritizing challenging tasks to optimize demands intensive and expensive policy evaluation in massive environments over iteration. To overcome the efficiency bottleneck, Wang et al. (2025) constructs a risk predictive model to actively infer MDP difficulties for worst subset selection in policy search, which we identify as a method of the robust active task sampling (RATS) paradigm in Fig. 1. As policy evaluation in arbitrary MDPs can be amortized by executing this risk predictive model, the resulting model predictive task sampling (MPTS) (Wang et al., 2025) enables expanding the scope of task subset selection, e.g., screening $\mathcal{B}$ from $\hat{\mathcal{B}}$ MDPs, to learn adaptive policies without extra environment interaction cost. This reflects RATS's huge potential for efficient, robust decision-making when exhaustive policy evaluation is prohibitive in the vast MDP space.

**Challenges in Theoretical Analysis and Implementations:** Despite RATS's promise in decision-making, we can still perceive several issues from its latest SOTA method MPTS. (i) No versatile tool is developed for theoretical analysis, e.g., the robustness concept in optimization, which is an indispensable consideration in risk-averse cases. (ii) It requires a dedicated pseudo batch size $\hat{\mathcal{B}}$ and other configurations. As implied in Meta-RL results (Wang et al., 2025), appropriately mixing up the random and predictive samplers is decisive in task subset selection; otherwise, it degrades generalization and robustness (See Sec. 3.2). (iii) There lack comprehensive discussions about acquisition

---
[*]Equal contribution  [1]Department of Automation, Tsinghua University, Beijing, China. Correspondence to: Xiangyang Ji <xyji@tsinghua.edu.cn>.

*Proceedings of the $42^{nd}$ International Conference on Machine Learning*, Vancouver, Canada. PMLR 267, 2025. Copyright 2025 by the author(s).

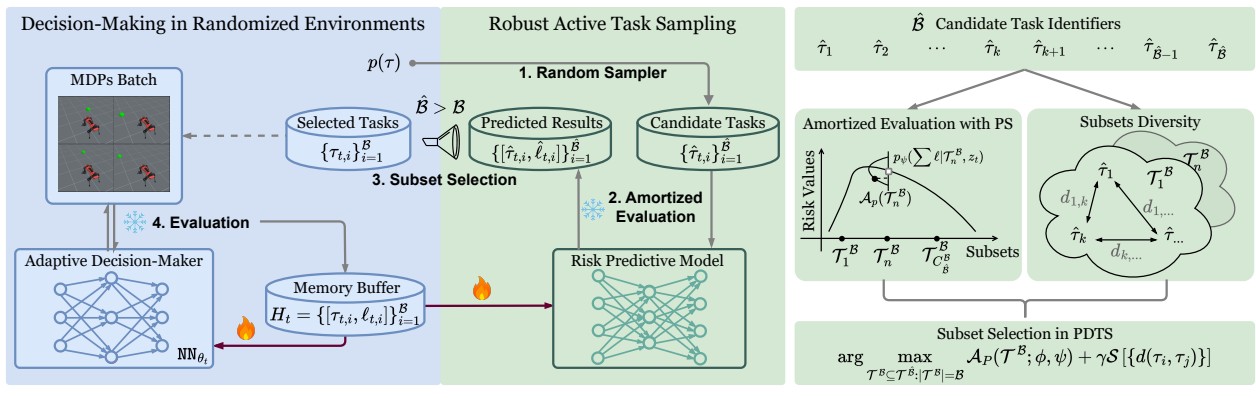

*Figure 1.* (a) General RATS in risk-averse decision-making. The pipeline involves amortized evaluation of task difficulties, robust subset selection, policy optimization in the MDP batch, and risk predictive models' update. [fire: updates; snow: evaluation] (b) PDTS as a RATS method. PDTS treats task subsets as bandit arms, evaluates values through posterior sampling, and solves a regularized problem.

principles in MPTS. The adopted upper confidence bound (UCB) principle must strike a balance between worst-case and uncertainty, with hyper-parameters carefully adjusted in implementation.

**Making Sense of RATS in Decision-Making:** Regarding theoretical understanding, we first abstract the general RATS process as a task-selection MDP $\mathcal{M}$, construct an infinitely many-armed bandit (i-MAB) (Carpentier & Valko, 2015) for task subset selection and demonstrate MPTS (Wang et al., 2025) as a special solution to the i-MAB. Aim at exploring more optimal subsets with the risk predictive model; we make a diagnosis of the concentration issue under a larger $\hat{\mathcal{B}}$ and enhance the acquisition function with the diversity regularization (Wang et al., 2023b; Borodin et al., 2017). To simplify the amortized evaluation and exploit the stochastic optimism in i-MAB, we adopt the posterior sampling strategy (Russo & Van Roy, 2014) to search for the optimal task subset with fewer configurations. Under these modifications, we present the Posterior and Diversity Synergized Task Sampling (PDTS) as a competitive RATS method.

**Contributions and Fascinating Discoveries in PDTS.** This work is built on empirical findings and the risk predictive model in MPTS, while the research focus is orthogonal (See Table 1). Surrounding risk-averse adaptive decision-making, the primary contributions are:

1. Our constructed i-MAB provides a versatile model to achieve RATS under various principles, including but not limited to MPTS and PDTS. The separate robustness concepts can be refined accordingly.
2. The designed diversity regularized acquisition function fixes the concentration issue, allows for exploration in a wider range of task sets (e.g., $\hat{\mathcal{B}} = 64\mathcal{B}$), and secures nearly worst-case MDP robustness.
3. The resulting PDTS is easy-to-implement and benefits

from stochastic optimism in posterior sampling for decision-making.

Empirically, the most thrilling finding is that PDTS exhibits adaptation robustness superior to existing SOTA baselines in typical DR and Meta-RL benchmarks without complicated configurations. Even in more realistic and challenging scenarios, such as vision-based decision-making, PDTS retains a remarkable performance over others.

## 2. Research Background

**Notations.** For conciseness and coherence, we retain most notations in (Wang et al., 2025) and leave the MDP distribution $p(\tau)$ details and RL preliminaries in Appendix A. Both DR and Meta-RL perform policy search in $p(\tau)$, where MDPs as tasks are specified by physics identifiers $\boldsymbol{\tau} \in \mathbb{R}^d$. The primary goal of DR (Tobin et al., 2017; Muratore et al., 2018) and Meta-RL (Beck et al., 2023) is to seek a policy $\theta \in \Theta$ that adapts well to a new MDP with zero or a few rollouts. We denote the task-specific dataset by $\mathcal{D}_\tau = \mathcal{D}_\tau^S \cup \mathcal{D}_\tau^Q$, where $\mathcal{D}_\tau^S$ are the $K$-rollouts for fast adaptation in MDP $\tau$ and $\mathcal{D}_\tau^Q$ are rollouts for after-adaptation policy evaluation. DR differs from Meta-RL and learns to adapt in a zero-shot manner (Mehta et al., 2020; Chi et al., 2024a), indicating $\mathcal{D}_\tau^S = \emptyset$. The risk function is $\ell : \mathcal{D}_\tau \times \Theta \mapsto \mathbb{R}$, mapping to the adaptation risk value, e.g., negative average returns of query rollouts.

In task episodic learning of DR and Meta-RL, the optimization history is written as $\hat{H}_t = \left\{ \boldsymbol{\theta}_t, \left( \boldsymbol{\tau}_{t,i}, \mathcal{D}_{\tau_{t,i}}, \ell_{t,i} \right) \right\}_{i=1}^{\mathcal{B}}$, with $\mathcal{B}$ the number of tasks to optimize in $t$-th iteration. MPTS (Wang et al., 2025) further prepares it as $H_t = \left\{ \left( \boldsymbol{\tau}_{t,i}, \mathcal{D}_{\tau_{t,i}}, \ell_{t,i} \right) \right\}_{i=1}^{\mathcal{B}}$ to feed into the risk learner to predict adaptation risk on arbitrary task. Importantly, it introduces the pseudo task set $\mathcal{T}_t^{\hat{\mathcal{B}}}$ with $\hat{\mathcal{B}} > \mathcal{B}$ and employ the acqui-

sition function $\mathcal{A}(\cdot)$ to actively select the subset for next iteration. Throughout this work, we leave the exact optimization task batch size $\mathcal{B}$ fixed and same for all methods in both theoretical analysis and evaluation.

## 2.1. Task Robust Optimization Methods

Robustness (Carlini et al., 2019; Chi et al., 2024b) is entangled with risk minimization principles. And this part recaps those incorporated in DR and Meta-RL for robust decision-making.

**Expected/Empirical Risk Minimization (ERM).** ERM originates from the statistical learning theory (Vapnik et al., 1998). Such a principle minimizes the expectation of risk values under $p(\tau)$:

$$\min_{\boldsymbol{\theta}\in\Theta} \mathbb{E}_{p(\tau)}\Big[\ell(\mathcal{D}_\tau^Q, \mathcal{D}_\tau^S; \boldsymbol{\theta})\Big], \qquad (1)$$

where $p(\tau)$ is mostly a uniform distribution as default.

**Group Distributionally Robust Risk Minimization (GDRM).** Such a principle aims to boost the adaptation robustness under certain proportional scenarios, e.g., a group of some under-sampled yet challenging tasks (Sagawa et al., 2019). In mathematics, we can write the expression as:

$$\min_{\boldsymbol{\theta}\in\Theta} \max_{g\in\mathcal{G}} \mathbb{E}_{p_g(\tau)}\Big[\ell(\mathcal{D}_\tau^Q, \mathcal{D}_\tau^S; \boldsymbol{\theta})\Big], \qquad (2)$$

with $\mathcal{G}$ to denote a collection of groups over a task dataset. GDRM handles extremely worst subpopulation shifts (Koh et al., 2021) through a risk-reweighting mechanism.

**Distributionally Robust Risk Minimization (DRM).** As a typical strategy in DRM, $\text{CVaR}_\alpha$ selects $(1-\alpha)$ proportional worst tasks after exact evaluation to optimize:

$$\min_{\theta\in\Theta} \text{CVaR}_\alpha(\boldsymbol{\theta}) := \mathbb{E}_{p_\alpha(\tau;\theta)}\Big[\ell(\mathcal{D}_\tau^Q, \mathcal{D}_\tau^S; \theta)\Big]. \quad (3)$$

In Meta-RL, this corresponds to the hard MDP prioritization (Greenberg et al., 2024; Lv et al., 2024). With $\min_{\theta\in\Theta}\lim_{\alpha\mapsto 1}\text{CVaR}_\alpha(\boldsymbol{\theta})$, it degenerates to the worst-case optimization in (Collins et al., 2020).

## 2.2. RATS Preliminaries & MPTS Modules

Here, we specify RATS paradigm, which slightly differs from traditional active learning purposes (Cohn et al., 1996). RATS is mainly incorporated into risk-averse learning with traits: (i) active inference towards task difficulties with limited cost, and (ii) acquisition rules to select the subset from the pseudo task set to optimize for robustness.

The adaptation capability in either few-shot (Wang et al., 2022; Chi et al., 2021) or zero-shot is our primary focus to evaluate in RATS. Here, MPTS designs a risk predictive model $p(\ell|\boldsymbol{\tau}, H_{1:t}; \boldsymbol{\theta}_t)$ to surrogate expensive evaluation, e.g., negative average rollout returns $\ell$ of the policy $\boldsymbol{\theta}_t$ in a MDP $\tau$. Hence, we identify it as a method of RATS. In particular, the empirical evidence in (Wang et al., 2025) Fig. 5 validates its risk predictive model's feasibility of approximately scoring MDPs' difficulties with high Pearson correlation coefficients between the model predictive ones and exact evaluation. Next, we overview its construction together with the acquisition function.

**Generative Modeling Adaptation Optimization.** MPTS treats the optimization history as sequence generation and involves latent variables $\boldsymbol{z}_t$ to summarize batches of adaptation risk over iterations, which leads to:

$$p(\mathcal{L}_{0:T}^{\mathcal{B}}, \boldsymbol{z}_{0:T}|\boldsymbol{\theta}_{0:T}) = p(\boldsymbol{z}_0)\prod_{t=0}^{T} p(\mathcal{L}_t^{\mathcal{B}}|\boldsymbol{z}_t, \boldsymbol{\theta}_t)\prod_{t=0}^{T-1} p(\boldsymbol{z}_{t+1}|\boldsymbol{z}_t), \tag{4}$$

with the evaluation risk batch $\mathcal{L}_t^{\mathcal{B}} = \{(\boldsymbol{\tau}_{t,i}, \ell_{t,i})\}_{i=1}^{\mathcal{B}}$.

With the Bayes rule and the streaming variational inference (Broderick et al., 2013; Nguyen et al., 2017) *w.r.t.* Eq. (4), it obtains the approximate evidence lower bound of the risk learner to maximize in each batch:

$$\max_{\boldsymbol{\psi}\in\Psi, \boldsymbol{\phi}\in\Phi} \mathcal{G}_{\text{ELBO}}(\boldsymbol{\psi}, \boldsymbol{\phi}) := \mathbb{E}_{q_{\boldsymbol{\phi}}(\boldsymbol{z}_t|H_t)}\left[\sum_{i=1}^{\mathcal{B}} \ln p_{\boldsymbol{\psi}}(\ell_{t,i}|\boldsymbol{\tau}_{t,i}, \boldsymbol{z}_t)\right]$$
$$- \beta D_{KL}\Big[q_{\boldsymbol{\phi}}(\boldsymbol{z}_t|H_t) \parallel q_{\bar{\boldsymbol{\phi}}}(\boldsymbol{z}_t|H_{t-1})\Big], \quad (5)$$

with $\bar{\boldsymbol{\phi}}$ the fixed conditioned prior from the last update and $\beta \in \mathbb{R}^+$ the penalty weight. Then the amortized evaluation of adaptation performance is approximated as $p(\ell|\boldsymbol{\tau}, H_{1:t}; \boldsymbol{\theta}_t) \approx \mathbb{E}_{q_{\boldsymbol{\phi}}(\boldsymbol{z}_t|H_t)}[p_{\boldsymbol{\psi}}(\ell|\boldsymbol{\tau}, \boldsymbol{z}_t; \boldsymbol{\theta}_t)]$ through Monte Carlo estimates.

$$\max_{\boldsymbol{\psi}\in\Psi} \mathcal{L}_{\text{ML}}(\boldsymbol{\psi}) := \ln p_{\boldsymbol{\psi}}(H_t|H_{1:t-1}) \quad (6a)$$

$$p(\ell|\hat{\boldsymbol{\tau}}_i, H_{1:t}; \boldsymbol{\theta}_t) \xrightarrow{\text{MC}} \{m(\ell_i), \sigma(\ell_i)\}_{i=1}^{\hat{\mathcal{B}}} \quad (6b)$$

$$\mathcal{T}_{t+1}^{\mathcal{B}*} = \arg\max_{\mathcal{T}_{t+1}^{\mathcal{B}}\subseteq\mathcal{T}_{t+1}^{\hat{\mathcal{B}}}:|\mathcal{T}_{t+1}^{\mathcal{B}}|=\mathcal{B}} \mathcal{A}_{\text{U}}(\mathcal{T}_{t+1}^{\mathcal{B}}) \quad (6c)$$

**Optimization Pipeline.** MPTS contains three critical steps in accordance with Eq. (6). ① In *approximate posterior inference* for Eq. (6)a, MPTS optimizes the risk predictive model through maximizing Eq. (5) with $H_t$; ② In *amortized evaluation*, it samples $\mathcal{T}_{t+1}^{\hat{\mathcal{B}}}$ from $p(\tau)$ and runs stochastic forward passes to estimate $m(\ell_i)$ and $\sigma(\ell_i)$ $\forall\hat{\tau} \in \mathcal{T}_t^{\hat{\mathcal{B}}}$ in Eq. (6)b; ③ In *subset selection*, it picks up the subset $\mathcal{T}_{t+1}^{\mathcal{B}}$ among Top-$\mathcal{B}$ acquisition scores from $\mathcal{T}_{t+1}^{\hat{\mathcal{B}}}$ for next iteration optimization. By repeating ①-③ steps for $T$-rounds, MPTS derives the robust adaptive decision-maker.

**Acquisition Function in MPTS.** The subset selection rule is built on the principle of optimism in the face of uncertainty (OFU) (Auer, 2002b), and tasks with worse adaptation performance and higher epistemic uncertainty are prioritized. This leads to the UCB principle, i.e., $\mathcal{A}_\mathrm{U}(\mathcal{T}_{t+1}^{\mathcal{B}})$ (Auer, 2002a; Garivier & Moulines, 2008):

$$\mathcal{A}_\mathrm{U}(\mathcal{T}_{t+1}^{\mathcal{B}}) = \sum_{i=1}^{\mathcal{B}} \gamma_0 m(\ell_i) + \gamma_1 \sigma(\ell_i), \text{ with } \tau_i \in \mathcal{T}_{t+1}^{\mathcal{B}}, \quad (7)$$

where the mean $m(\ell_i) = \mathbb{E}_{q_\phi(\boldsymbol{z}_t|H_t)}\left[p_\psi(\ell|\boldsymbol{\tau}_i, \boldsymbol{z}_t)\right]$ and the standard deviation $\sigma(\ell_i) = \mathbb{V}_{q_\phi(\boldsymbol{z}_t|H_t)}^{\frac{1}{2}}\left[p_\psi(\ell|\boldsymbol{\tau}_i, \boldsymbol{z}_t)\right]$ of task-specific adaptation risk and are estimated through multiple stochastic forward passes, i.e., $\boldsymbol{z}_t \sim q_\phi(\boldsymbol{z}_t|H_t)$ and $\ell_i \sim p_\psi(\ell_i|\boldsymbol{\tau}_i, \boldsymbol{z}_t)$. $\{\gamma_0, \gamma_1\}$ are trade-off parameters.

## 3. Theoretical Investigations and Practical Enhancements

This section first studies RATS with a task-selection MDP, proposes i-MABs as a versatile tool for RATS methods and establishes its connections with MPTS. To promote RATS's use in risk-averse decision-making, we analyze acquisition rules' influence on robustness concepts and present PDTS as an easy-to-implement yet powerful scheme.

### 3.1. Enable Robust Active Task Sampling with i-MABs

When RATS meets decision-making, it involves scoring MDPs' difficulty from amortized evaluation, selecting a subset, and performing adaptive policy search in either a zero-shot or few-shot manner. The following introduces a theoretical tool to analyze these steps in RATS.

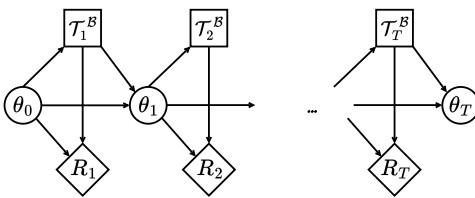

*Figure 2.* **Task Robust Episodic Learning as a Task-Selection MDP.**

**Task Robust Episodic Learning as a MDP.** The primary insight lies in a finite-horizon Markov decision process (Puterman, 2014), denoted by $\mathcal{M} = <\mathbf{S}, \mathbf{A}, \mathbf{P}, \mathbf{R}>$. Here, we specify the essential components of $\mathcal{M}$ as:

- *State Space.* $\mathcal{M}$ treats the feasible machine learner's parameter, such as Meta-RL policies, as the reachable state, i.e., $\mathbf{S} = \{\boldsymbol{\theta} \in \boldsymbol{\Theta}\}$;
- *Action Space.* The action space constitutes a collection of task subsets with cardinality constraints $\mathbf{A}_t = \{\mathcal{T}_t^{\mathcal{B}} \subseteq \mathcal{T}_t^{\hat{\mathcal{B}}} \text{ with } |\mathcal{T}_t^{\mathcal{B}}| = \mathcal{B}\}$ in RATS or $\mathrm{CVaR}_\alpha$ methods;

- *Transition Dynamics.* We describe the dynamical system as $p(\boldsymbol{\theta}_{t+1}|\boldsymbol{\theta}_t, \mathcal{T}_{t+1}^{\mathcal{B}}) \in \mathbf{P}$ conditioned on $\boldsymbol{\theta}_t$ and the action $\mathcal{T}_{t+1}^{\mathcal{B}}$ with the transited state (after-adaptation) $\boldsymbol{\theta}_{t+1}$;
- *Reward Function.* The step-wise reward quantifies adaptation robustness improvement after state transitions. With $\mathrm{CVaR}_\alpha$ as a risk-averse measure, we define it as $R(\boldsymbol{\theta}_t, \mathcal{T}_{t+1}^{\mathcal{B}}) := \mathrm{CVaR}_\alpha(\boldsymbol{\theta}_t) - \mathrm{CVaR}_\alpha(\boldsymbol{\theta}_{t+1})$.

Given $\mathcal{M}$ in Fig. 2, the maximum iteration step $T$ also corresponds to the total interaction rounds. The agent actively selects $\mathcal{T}_{t+1}^{\mathcal{B}}$ from the time-varying $\mathcal{T}_{t+1}^{\hat{\mathcal{B}}}$ and optimize the machine learner to increase adaptation robustness in $T$-rounds.

Accordingly, the sequential actions are the outcome of a series of deterministic functions as the policy set $\Pi_{0:T-1} = \{\pi_t\}_{t=0}^{T-1}$. Here, the policy maps $H_{0:t}$ to the next subset from $\mathcal{T}_{t+1}^{\hat{\mathcal{B}}}$ for optimization, i.e., $\pi_t : H_{0:t} \mapsto \mathcal{T}_{t+1}^{\mathcal{B}}$. These ingredients can depict $\mathcal{M}$ in a probabilistic graph as Fig. 2.

Formally, we can express the agent's ultimate goal as maximizing the cumulative reward in a sequential manner,

$$\Pi_{0:T-1}^* = \arg\max_{\Pi_{0:T-1}} \sum_{t=0}^{T-1} R(\boldsymbol{\theta}_t, \mathcal{T}_{t+1}^{\mathcal{B}}). \quad (8)$$

Note that solving Eq. (8) is equivalent to reaching the optimal state $\boldsymbol{\theta}_T^*$ with lowest $\mathrm{CVaR}_\alpha(\boldsymbol{\theta})$ after repeating $T$-round optimization steps.

We also denote the policy subset of an arbitrary intermediate decision-making sequence by $\Pi_{k:j} := \{\pi_i\}_{i=k}^{j}$ with its optimal solution marked in $^*$. The associated cumulated return is $\sum_{i=k}^{j} R(\boldsymbol{\theta}_i, \mathcal{T}_{i+1}^{\mathcal{B}*})$ conditioned on the starting state $\boldsymbol{\theta}_k$.

*Remark* 3.1 (Bellman Optimality). For the studied $T$-horizon $\mathcal{M}$, we write its Bellman optimality as:

$$\sum_{i=0}^{T-1} R(\boldsymbol{\theta}_i, \mathcal{T}_{i+1}^{\mathcal{B}*}) = \max_{\Pi_{0:t-1}} \sum_{i=0}^{t-1} R(\boldsymbol{\theta}_i, \mathcal{T}_{i+1}^{\mathcal{B}}) + \sum_{i=t}^{T-1} R(\boldsymbol{\theta}_i, \mathcal{T}_{i+1}^{\mathcal{B}*}),$$
$$(9)$$

revealing that the optimal solution to the sub-problem also reserves its global optimality. Hence, we can break the problem-solving into optimal subset selection in each round.

**From i-MABs to MPTS's Robustness Concept.** Finding plausible strategies to solve $\mathcal{M}$ is non-trivial as policy search is considered in a discrete space with the varying action set $\mathbf{A}_t$. To this end, we further simplify $\mathcal{M}$ into an infinite MAB (Mahajan & Teneketzis, 2008) to enable an online search of the optimal subset and interpret the optimization pipeline of MPTS as a special case.

Distinguished from the vanilla multi-armed bandit, $\mathcal{M}$ involves the state $\boldsymbol{\theta}_t$ and $\mathbf{A}_{t+1}$ induced by resampled $\mathcal{T}_{t+1}^{\hat{\mathcal{B}}}$. The arm corresponds to a feasible subset $\mathcal{T}_{t+1}^{\mathcal{B}} \in \mathbf{A}_{t+1}$. Hence, we can associate the reward distribution $p(R|\boldsymbol{\theta}_t, \mathcal{T}_{t+1}^{\mathcal{B}})$ with the chosen action $\mathcal{T}_{t+1}^{\mathcal{B}}$. As a result, one

*Table 1.* Comparison between MPTS and PDTS in Contributions.

| | Research Lens | Robustness Concept | Subset Selection | Acquisition Rule |
|---|---|---|---|---|
| MPTS (Wang et al., 2025) | Generative Modeling | Nearly CVaR$_\alpha$ | Max-Sum (Top-$\mathcal{B}$) | UCB |
| PDTS (Ours) | MDP & i-MABs | Nearly Worst-Case | Max-Sum Diversity | Posterior Sampling |

optimistic strategy for i-MABs is to execute greedy search $\mathcal{T}_{t+1}^{\mathcal{B}*} = \arg\max_{\mathcal{T}_{t+1}^{\mathcal{B}} \in \mathbf{A}_{t+1}} \mathbb{E}\left[R|\boldsymbol{\theta}_t, \mathcal{T}_{t+1}^{\mathcal{B}}\right]$ in each round.

The regret in the i-MAB measures the performance difference between the cumulated rewards of the optimal policy $\Pi_{0:T-1}^*$ and the actually executed policy $\Pi_{0:T-1}$ over $T$-rounds:

$$\text{Regret}(T, \Pi_{0:T-1}) = \sum_{t=0}^{T-1} \left[R(\boldsymbol{\theta}_t^*, \mathcal{T}_{t+1}^{\mathcal{B}*}) - R(\boldsymbol{\theta}_t, \mathcal{T}_{t+1}^{\mathcal{B}})\right]. \tag{10}$$

We can further simplify the definition expression as $\text{Regret}(T, \Pi_{0:T-1}) = \text{CVaR}_\alpha(\boldsymbol{\theta}_T) - \text{CVaR}_\alpha(\boldsymbol{\theta}_T^*)$, which quantifies the performance gap between the optimal state and the practical state under $(1-\alpha)$-tail robustness.

**Proposition 3.2** (MPTS as a UCB-guided Solution to i-MABs). *Executing MPTS pipeline in Eq. (6) is equivalent to approximately solving $\mathcal{M}$ with the i-MAB under the UCB principle.*

Consequently, our i-MAB is a theoretical model for inducing RATS methods, and MPTS can be viewed as a special case in **Proposition** 3.2.

### 3.2. Acquisition Functions Matter in Improving Coverage & Boosting Robustness

Several decision-making scenarios, e.g., robotics, are risk-averse, which makes worst-case optimization more advantageous (Greenberg et al., 2024). However, (i) the search scope to worst cases is restricted by the batch size, and (ii) minimax optimization requires carefully designed relaxation in the field (Collins et al., 2020; Sagawa et al., 2019).

**Benefits of Enlarging $\hat{\mathcal{B}}$ in Subset Selection.** Note that the feasible subset is the arm in the i-MAB, enlarging $\hat{\mathcal{B}}$ increases its number to $|\mathbf{A}_t| = C_{\hat{\mathcal{B}}}^{\mathcal{B}}$. One promising trait of the risk predictive model in MPTS is to amortize the policy evaluation in arbitrary MDP without exact interactions. Hence, greater $\hat{\mathcal{B}}$ in implementation reserves at least two bonus: (i) it encourages exploration in the task space with more candidate subsets at no actual interaction cost; (ii) under high-risk prioritization rule, the optimization pipeline in RATS approximately executes $\text{CVaR}_{1-\frac{\mathcal{B}}{\hat{\mathcal{B}}}}$ in each iteration, i.e., worst-case optimization with $\hat{\mathcal{B}} \to \infty$. Notably, the additional computational overhead with larger $\hat{\mathcal{B}}$ remains negligible due to the efficiency of the risk predictive model—its cost is significantly lower than that of agent-environment

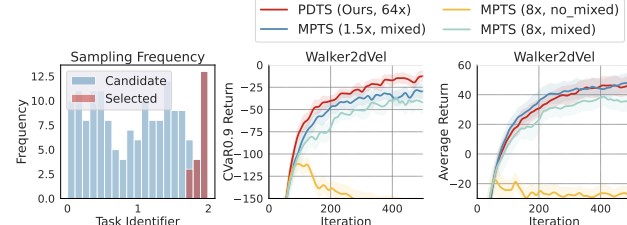

*Figure 3.* **MPTS's Performance Collapse with Greater $\hat{\mathcal{B}}$.** We report the performance collapses of MPTS on Walker2dVel in the case $\hat{\mathcal{B}} = 8\mathcal{B}$. The task sampling frequency reveals the presence of the concentration issue.

interactions and policy optimization in Meta-RL or DR.

Unfortunately, MPTS might encounter performance collapse with greater $\hat{\mathcal{B}}$, as indicated in Fig. 3 when $\hat{\mathcal{B}} = 8\mathcal{B}$ without a remedy. This empirical result can be attributed to the selected subset's concentration in a narrow range, which motivates the proposal of heuristic tricks in Meta-RL to adopt a mixed sampling strategy to ensure sufficient visitations to the whole task space (Wang et al., 2025; Greenberg et al., 2024). Although these heuristics mitigate the issue, they do not provide a complete solution, as performance degradation persists with increasing $\mathcal{B}$.

**Theoretical Diagnosis of Sealed Exploration Potential in MPTS.** In practice, we propose to circumvent complicated heuristics and retain the simplicity to enable sufficient exploration in i-MABs. To this end, we analyze the concentration issue and report the theoretical analysis as **Proposition** 3.3.

**Proposition 3.3** (Concentration Issue in Average Top-$\mathcal{B}$ Selection). *Let $f(\boldsymbol{\tau}) : \mathbb{R}^d \to \mathbb{R}$ be a unimodal and continuous function, where $d \in \mathbb{N}^+$ and $\boldsymbol{\tau} \in \mathbb{R}^d$, with a maximum value $f(\boldsymbol{\tau}^*)$ at $\boldsymbol{\tau}^*$. We uniformly sample a set of points $\mathcal{T}^{\hat{\mathcal{B}}} = \{\boldsymbol{\tau}_i\}_{i=1}^{\hat{\mathcal{B}}}$, where $\boldsymbol{\tau}_i$ are i.i.d. with a probability $p_\epsilon$ of falling within a $\epsilon$-neighborhood of $\boldsymbol{\tau}^*$ as $|f(\boldsymbol{\tau}) - f(\boldsymbol{\tau}^*)| \le \epsilon$. Following MPTS, we select the Top-$\mathcal{B}$ samples with the largest function values, i.e.,*

$$\mathcal{T}^{\mathcal{B}} = \text{Top-}\mathcal{B}(\mathcal{T}^{\hat{\mathcal{B}}}, f), \quad \hat{\mathcal{B}}, \mathcal{B} \in \mathbb{N}^+, \ \mathcal{B} \le \hat{\mathcal{B}},$$

*For any $\epsilon > 0$ such that $p_\epsilon < \frac{\hat{\mathcal{B}} - \mathcal{B} + 2}{\hat{\mathcal{B}} + 1}$, the concentration probability*

$$\mathbb{P}\left(|f(\boldsymbol{\tau}) - f(\boldsymbol{\tau}^*)| \le \epsilon \mid \forall \boldsymbol{\tau} \in \mathcal{T}^{\mathcal{B}}\right)$$

*increases with $\hat{\mathcal{B}}$ and converges to 1 with $\hat{\mathcal{B}} \to \infty$.*

As implied, with the increase of $\hat{\mathcal{B}}$ and fixed $\mathcal{B}$, the Top-$\mathcal{B}$ operator tends to select the subset in a small neighborhood of $\arg\max_{\boldsymbol{\tau}} f(\boldsymbol{\tau})$, which corresponds to the subset concentration issue in MPTS, over-optimizes a local region and hampers task space exploration. Due to the use of Top-$\mathcal{B}$ operator in batch optimization, we also hypothesize a similar phenomenon in DRM (Wang et al., 2024c; Lv et al., 2024). Next, we will propose a natural plausible mechanism to enlarge $\hat{\mathcal{B}}$ in RATS without suffering concentration pains.

**Diversity Regularization & Robustness Concept.** Our strategy is to encourage the coverage of the task space during subset selection. Specifically, rather than selecting individual candidate tasks based on their acquisition score, we evaluate task batches based on both adaptation risk and task diversities in the subset. This formulation constructs a diversity maximization problem (Wang et al., 2023b):

$$\max_{\mathcal{T}^{\mathcal{B}} \subseteq \mathcal{T}^{\hat{\mathcal{B}}} : |\mathcal{T}^{\mathcal{B}}| = \mathcal{B}} \mathcal{A}(\mathcal{T}^{\mathcal{B}}) + \gamma \mathcal{S}\left[\{d(\boldsymbol{\tau}_i, \boldsymbol{\tau}_j)\}\right] \qquad (11)$$

where $\mathcal{S}$ measures the diversity of the subset $\mathcal{T}^{\mathcal{B}}$ from identifiers' pairwise distances, e.g., $\sum_{i,j} ||\boldsymbol{\tau}_i - \boldsymbol{\tau}_j||_2^2$.

Though the involvement of the regularization term makes the subset selection problem NP-hard, it can be solved using simple approximate algorithms (Borodin et al., 2017; Wang et al., 2023b).

**Proposition 3.4** (Nearly Worst-Case Optimization with PDTS). *When $\hat{\mathcal{B}}$ grows large enough, optimizing the subset from Eq. (11) achieves nearly worst-case optimization.*

**Proposition** 3.4 indicates that the regularized acquisition rule in Eq. (11) enables exploration of the worst subset from numerous candidate arms while retaining the subset's coverage of the task space to a certain level. Compared with (Collins et al., 2020; Sagawa et al., 2019), such a regularization will show its effectiveness in experiments.

**3.3. Practical Sampling with Stochastic Optimism**

So far, we have presented the i-MAB as a theoretical tool for RATS and interpreted MPTS as the UCB-guided solution. However, the UCB acquisition rule requires multiple stochastic forward passes for each task's evaluation, and computations grow with $\hat{\mathcal{B}}$. It also demands calibration of exploration and exploitation weights in subset search.

To cut off unnecessary computations and retain the uncertainty optimism, we adopt the posterior sampling strategy as the acquisition principle for RATS. The posterior sampling (Osband et al., 2013; Asmuth et al., 2012) is an extension of Thompson sampling (Thompson, 1933) in solving MDPs with infinite actions. The reward or action value is treated as a randomized function, and each arm's value, sampled from the posterior once, serves action selection, i.e., $\mathcal{A}_{\text{P}}(\mathcal{T}^{\mathcal{B}}) = \sum_{i=1}^{\mathcal{B}} \hat{\ell}_{t+1,i}$ in Eq. (12).

---

$$\text{One Forward Pass:} \quad \boldsymbol{z}_t \sim q_{\boldsymbol{\phi}}(\boldsymbol{z}_t | H_t) \quad (12\text{a})$$

$$\hat{\ell}_{t+1,i} \sim p_{\boldsymbol{\psi}}(\ell | \hat{\boldsymbol{\tau}}_i, \boldsymbol{z}_t) \quad \forall i \in \{1, \dots, \hat{\mathcal{B}}\} \quad (12\text{b})$$

$$\mathcal{T}_{t+1}^{\mathcal{B}*} = \arg \max_{\substack{\mathcal{T}^{\mathcal{B}} \subseteq \mathcal{T}^{\hat{\mathcal{B}}} \\ |\mathcal{T}^{\mathcal{B}}| = \mathcal{B}}} \mathcal{A}_{\text{P}}(\mathcal{T}^{\mathcal{B}}) + \gamma \mathcal{S}\left[\{d(\boldsymbol{\tau}_i, \boldsymbol{\tau}_j)\}\right] \quad (12\text{c})$$

---

As implied in (Russo & Van Roy, 2014), posterior sampling avoids over-exploitation of inaccurate estimated uncertainty, and benefits more from stochasticity. Together with diversity regularization, we obtain PDTS as a new RATS method in Eq. (12). Intuitively, diversity penalty and posterior sampling suppress inaccurate Top-$\mathcal{B}$ operations and synergize exploration in the task space. PDTS enjoys implementation simplicity, computational efficiency and stochastic optimism in decision-making.

**3.4. Overall Implementation**

Putting the above modules together, we present PDTS in Algorithm 1, which is easily wrapped into DR and Meta-RL (See Algorithm 2 and 4).

---

**Algorithm 1** Posterior-Diversity Synergized Task Sampling

**Input** : Task distribution $p(\boldsymbol{\tau})$; Task batch size $\mathcal{B}$; Candidate batch size $\hat{\mathcal{B}}$; Latest updated $\{\boldsymbol{\psi}, \boldsymbol{\phi}\}$; Latest history $H_{t-1}$; Iteration number $K$; Learning rate $\lambda_2$.

**Output** : Selected task identifier batch $\{\boldsymbol{\tau}_{t,i}\}_{i=1}^{\mathcal{B}}$.

// **Optimize Risk Predictive Module**
**for** $i = 1$ **to** $K$ **do**
  Perform gradient updates given $H_{t-1}$:
    $\boldsymbol{\phi} \leftarrow \boldsymbol{\phi} + \lambda_2 \nabla_{\boldsymbol{\phi}} \mathcal{G}_{\text{ELBO}}(\boldsymbol{\psi}, \boldsymbol{\phi})$ in Eq. (5);
    $\boldsymbol{\psi} \leftarrow \boldsymbol{\psi} + \lambda_2 \nabla_{\boldsymbol{\psi}} \mathcal{G}_{\text{ELBO}}(\boldsymbol{\psi}, \boldsymbol{\phi})$ in Eq. (5);
**end**
// **Simulating After-Adaptation Results**
Randomly sample $\{\hat{\boldsymbol{\tau}}_{t,i}\}_{i=1}^{\hat{\mathcal{B}}}$ from $p(\boldsymbol{\tau})$;
// **Posterior Sampling Outcome**
Amortized evaluation $\{\hat{\ell}_{t,i}\}_{i=1}^{\hat{\mathcal{B}}}$ with one stochastic forward pass for all candidate tasks through executing Eq. (12)a-b;
// **Diversity-Guided Subset Search**
Run approximate algorithms to solve Eq. (12)c;
Return the screened subset $\{\boldsymbol{\tau}_{t,i}\}_{i=1}^{\mathcal{B}}$ for next iteration.

---

## 4. Experiments

This section presents experimental results on Meta-RL and robotics DR involving randomized physics properties.

**Risk-Averse Baselines.** For fair comparison, we retain the standard setup in (Wang et al., 2025) and use SOTA baselines as described in Section 2.1: ERM (Vapnik et al., 1998), DRM (Wang et al., 2024c; Greenberg et al., 2024),

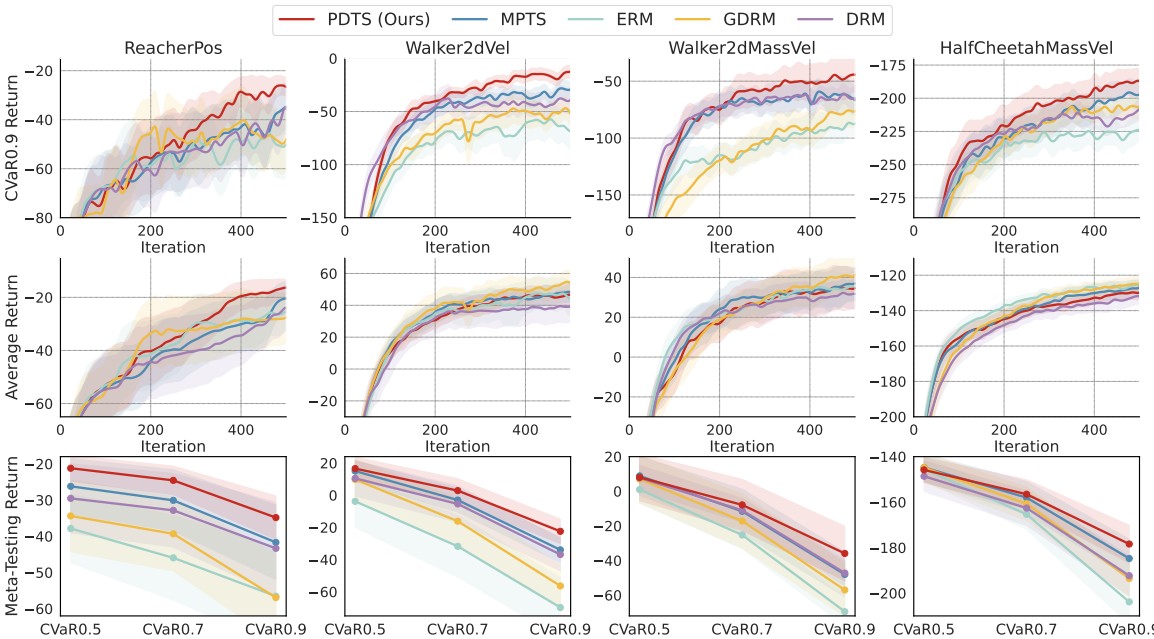

*Figure 4.* **Meta-RL Results.** The top depicts the cumulative return curves for CVaR$_{0.9}$ validation MDPs during meta-training; the middle shows the average cumulative returns curves during meta-training; and the bottom presents the meta-testing results with various $\alpha$.

GDRM (Sagawa et al., 2019), and MPTS. Following implementations in (Wang et al., 2025; Tao et al., 2024), we use MAML (Finn et al., 2017) as the backbone algorithm in Meta-RL and employ TD3 (Fujimoto et al., 2018) and PPO (Schulman et al., 2017) for domain randomization, respectively.

We run each algorithm on seven random seeds and report the average performance with standard error of means. For adaptation robustness, we compute CVaR$_\alpha$ values across the validation tasks, with $\alpha = \{0.9, 0.7, 0.5\}$, and also report some out-of-distribution (OOD) results. Importantly, the pseudo batch size is set to $\hat{\mathcal{B}} = 64\mathcal{B}$ as default for PDTS.

### 4.1. Meta Reinforcement Learning

We consider Meta-RL continuous control scenarios based on MuJoCo (Todorov et al., 2012): Reacher, Walker2d, and HalfCheetah. These include identifiers: target position, velocity, and mass (Wang et al., 2025).

Fig. 4 reveals PDTS consistent superiority over others in CVaR$_{0.9}$ validation return, demonstrating excellent adaptation robustness. Owing to intrinsic sampling randomness, PDTS and MPTS achieve average performance comparable to ERM, while DRM struggles to improve robustness and sacrifices more average returns. Surprisingly, benefiting from exploration, PDTS boosts both adaptation robustness and task efficiency on ReacherPos. We will further discuss this in Section 4.2. Regarding meta-testing, PDTS excels across CVaR values, consistent with the learning curves. No-

tably, PDTS's performance advantage increases with higher $\alpha$ values. In the extreme scenario CVaR$_{0.9}$, PDTS outperforms others by more than 15% on all benchmarks except HalfCheetahMassVel.

**PDTS is agnostic to the meta-learning backbone.** PDTS is a plug-and-play module agnostic to the meta-learning backbone. Here, we integrate PDTS with PEARL (Rakelly et al., 2019) and compare it with MPTS and RoML (Greenberg et al., 2024), built on the same backbone. As shown in Table 2, we evaluate these methods on two Meta-RL scenarios from RoML. PDTS outperforms others in terms of CVaR return, further examining its effectiveness and scalability in risk-averse decision-making.

*Table 2.* PDTS's compatibility with PEARL backbone. Baselines are MPTS and RoML (Greenberg et al., 2024).

| CVaR$_{0.95}$ Return | PDTS | MPTS | RoML | PEARL |
|---|---|---|---|---|
| HalfCheetahBody | **993±26** | 945±26 | 855±35 | 847±42 |
| HalfCheetahMass | **1296±41** | 1209±45 | 1197±59 | 1118±51 |

### 4.2. Physical Robotics Domain Randomization

We evaluate PDTS on three DR scenarios introduced by Mehta et al. (2020): a game scenario, LunarLander, and two robotic arm control scenarios, Pusher and ErgoReacher.

Empirical findings in Fig. 5(a) are consistent with those in Meta-RL, where PDTS dominates the robustness performance. Notably, PDTS's advantage is more pronounced on Pusher and LunarLander, where the identifiers' dimension is lower than ErgoReacher's. This is likely because low-dimensional task identifiers exacerbate the concentration

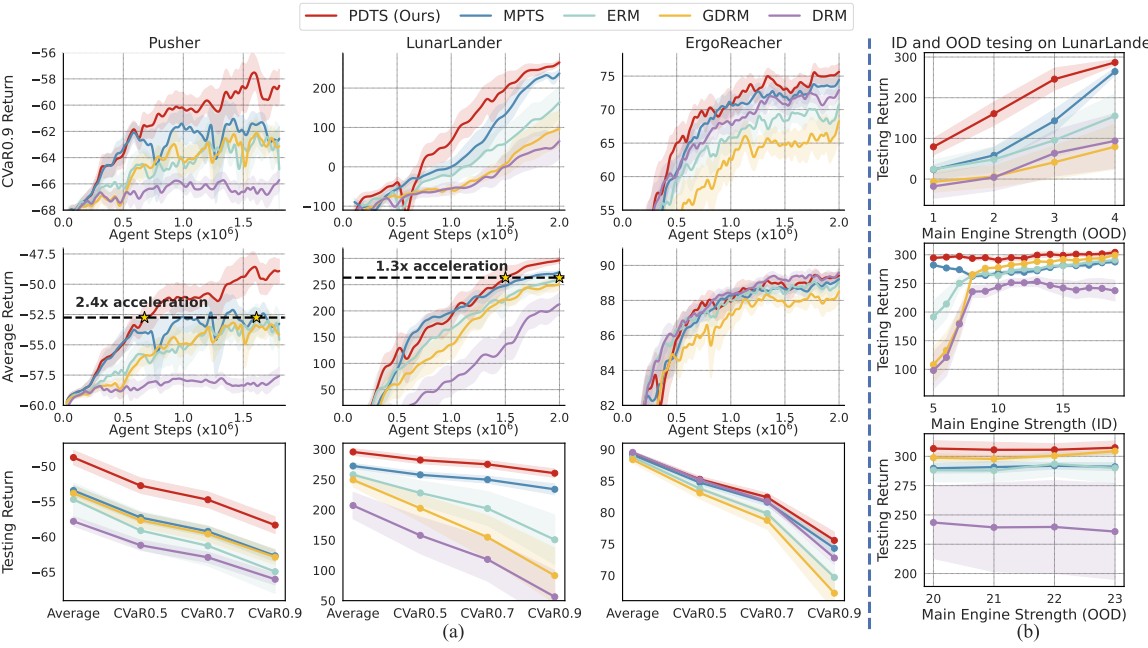

*Figure 5.* **Physical Robotics DR Results**. (a) The top shows the cumulative return curves for $CVaR_{0.9}$ validation MDPs during training; the middle displays the average cumulative return curves across all validation MDPs during training; and the bottom presents the test results at various $CVaR_{\alpha}$. (b) We evaluate the trained policies in both in-distribution (ID) and out-of-distribution (OOD) domains on LunarLander, reporting the average returns for each sampled task.

issue, limiting the performance of MPTS. GDRM and DRM perform poorly in both average performance and robustness. Given a fixed batch size, this weakness may stem from their limited exploration capacity. Regarding final testing performance, PDTS excels across CVaR values. Specifically, PDTS outperforms ERM by more than 8% on all benchmarks in $CVaR_{0.9}$, and by as much as 73% on LunarLander.

**PDTS improves task efficiency in specific scenarios.** Beyond adaptation robustness, PDTS shows the potential to accelerate training in specific scenarios. As shown in Fig. 5(a), PDTS achieves average returns comparable to ERM's best performance but with fewer training steps, achieving $2.4\times$ acceleration in Pusher and $1.3\times$ in LunarLander. We attribute this acceleration to the efficient exploration of acquisition criteria. Hence, PDTS holds promise for achieving robustness with reduced training steps in broader scenarios.

**PDTS achieves superior zero-shot policy adaptation in OOD MDPs.** On Lundarlander, Fig. 5(b) witnesses PDTS's highest average test returns across sampled tasks. To evaluate zero-shot adaptation in OOD MDPs, we shift the identifier interval $\tau \in [4.0, 20.0]$ to the OOD range $\tau \in [1.0, 4.0) \cup (20.0, 23.0]$. Notably, PDTS exhibits the smallest performance degradation, particularly in the most challenging OOD tasks ($\tau \in [1.0, 4.0)$).

## 4.3. Visual Robotics Domain Randomization

Vision-based robotics control is more challenging and underexplored in MPTS. With the latest robotics simulator, ManiSkill3 (Tao et al., 2024), we design two visual DR scenarios: one randomizing lighting with a table-top two-finger gripper arm robot called LiftPegUpright_Light, and another randomizing goal locations with a quadruped robot called AnymalCReach_Goal, as illustrated in Fig. 6(a).

PDTS achieves best robustness performance in Fig. 6(b), same as symbolic scenarios. Moreover, PDTS also exhibits a trend of accelerated training in terms of average performance, highlighting its potential for realistic scenarios requiring both fast and robust adaptation. Due to insufficient task exploration, DRM performs poorly and probably gets trapped in challenging tasks. ERM obtains the worst final performance on AnymalCReach_Goal. These findings underscore the importance of robust optimization with sufficient exploration, which may explain PDTS's success.

**PDTS can well discriminate task difficulties.** We evaluate the Pearson Correlation Coefficient (PCC) between the predicted episode returns and the exact values during training. As shown in Fig. 6(c), PDTS inherits MDPs' difficulty scoring capability of MPTS, with both PCC values greater than 0.5. In particular, PDTS exceeds MPTS in predicting accuracies, which may be attributed to the extended exploration and the stochastic optimism in posterior sampling.

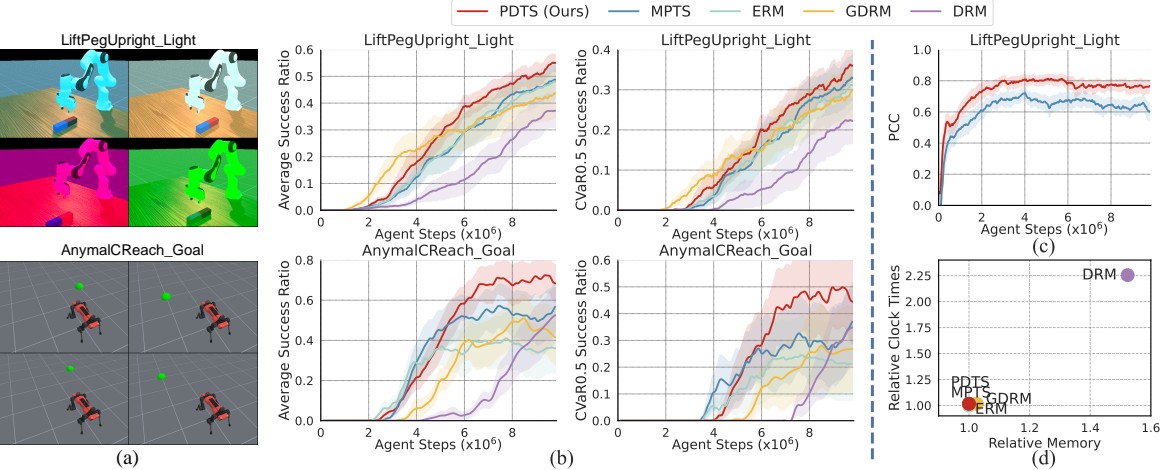

*Figure 6.* **Visual Robotics DR results.** (a) Illustrations of two visual DR scenarios. (b) Curves of the average success ratio and the $CVaR_{0.5}$ success ratio on validation tasks during training. (c) Training curves of PCC values between predicted and true episode returns. (d) Memory cost and clock time relative to ERM during meta-training.

**PDTS offers learning efficiency advantages at minimal cost.** Like MPTS, PDTS avoids additional evaluation costs associated with interaction and rendering, making it more beneficial in vision-based RL tasks. From relative clock time and memory usage in Fig. 6(d), we find PDTS retains computational efficiency with others except DRM. As a result, PDTS is a robust and scalable approach for complex, interaction-intensive tasks.

## 5. Conclusion

**Technical Discussions.** This work studies RATS's use for learning adaptive decision-makers in risk-averse decision-making. We present the i-MAB as a theoretical tool to enable RATS and establish theoretical connections with its latest method (Wang et al., 2025). Built on these insights, we propose PDTS as a competitive RATS method to achieve nearly worst-case optimization. Extensive DR and Meta-RL experiments validate the effectiveness of RATS and show its efficiency potential in risk-averse scenarios.

**Limitations & Extensions.** Our approach relies on the risk predictive model for roughly scoring task difficulties and leverages identifier information alongside the inherent smoothness of the adaptation risk function, though these assumptions may not always hold in restricted scenarios.

While robust active learning remains underexplored in sequential decision-making, reducing computational and sample expenses and improving robustness are crucial for building large-scale decision models. The models and algorithms introduced in this work offer a tractable strategy with strong empirical performance toward achieving RATS goals. Future explorations can include designing more accurate risk-predictive models to enhance amortized evaluation relia-

bility, as well as integrating stronger robust optimization techniques into i-MAB frameworks to develop practical and scalable RATS methods.

## Impact Statement

This paper presents work whose goal is to advance the field of Machine Learning. There are many potential societal consequences of our work, including enhancing real-world applications of generalized robotics and other decision-making systems. Although this study significantly improves adaptation robustness, adaptation failures remain possible, potentially leading to serious consequences in real-world scenarios. Thus, human supervision remains essential.

## Acknowledgments and Disclosure of Funding

This work is funded by National Natural Science Foundation of China (NSFC) with the Number # 62306326 and the National Key R&D Program of China under Grant 2018AAA0102801. We thank all reviewers for their positive comments and constructive suggestions in this work.

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

# A. Adaptive Decision-Making in Randomized Environments

## A.1. Zero-Shot and Few-Shot Sequential Decision-Making

In this part, we include typical MDPs and distributions, which align with randomized environments for sequential decision-making. Meanwhile, we show examples of DR and Meta-RL in this context, which can well specify the form of the task-specific adaptation risk $\ell$ in zero-shot and a few-shot setup.

**Definition A.1** (Identifier-Induced MDP Distribution). *We consider a distribution over MDPs $p(\mathcal{M}_\tau)$ induced by the identifier distribution $p(\tau)$. Here, the MDP as the sequential decision-making environment can be characterized as $\mathcal{M}_\tau =< \mathbf{S}, \mathbf{A}, \mathbf{P}_\tau, \mathbf{R}_\tau, \gamma >$, where $\mathbf{S}$ and $\mathbf{A}$ are the state and the action space shared across all MDPs.*

*MDPs vary in either the transition dynamics $\mathbf{P}_\tau := p_\tau(\boldsymbol{s}_{t+1}|\boldsymbol{s}_t, \boldsymbol{a}_t)$ or the reward function $\mathbf{R}_\tau := R_\tau(\boldsymbol{s}_t, \boldsymbol{a}_t)$ or the both, specified by corresponding identifiers $\boldsymbol{\tau}$. The $H$-horizon rollout under a policy is $(\boldsymbol{s}_0, \boldsymbol{a}_0, R(\boldsymbol{s}_0, \boldsymbol{a}_0), \ldots, \boldsymbol{s}_{H-1}, \boldsymbol{a}_{H-1}, R(\boldsymbol{s}_{H-1}, \boldsymbol{a}_{H-1}), \boldsymbol{s}_H)$ with its cumulative reward $\sum_{t=0}^{H-1} \gamma^t R(\boldsymbol{s}_t, \boldsymbol{a}_t)$ and $\gamma \in \mathbb{R}^+$.*

With Definition A.1, we can provide some instantiations of DR and Meta-RL under specific policy optimization backbones.

**Meta-RL with Model Agnostic Meta Learning.** Meta-RL seeks to train agents capable of rapidly adapting to new tasks or environments by utilizing knowledge from related tasks. We adopt MAML (Finn et al., 2017) as the standard backbone algorithm due to its widespread application in addressing few-shot sequential decision-making problems. This also retains the same configuration in MPTS (Wang et al., 2025). The optimization objective of MAML is mathematically defined as:

$$\min_{\boldsymbol{\theta} \in \boldsymbol{\Theta}} \mathbb{E}_{p(\tau)} \left[ \ell \left( \mathcal{D}_\tau^Q; \boldsymbol{\theta} - \lambda \nabla_{\boldsymbol{\theta}} \ell \left( \mathcal{D}_\tau^S; \boldsymbol{\theta} \right) \right) \right] \tag{13a}$$

$$\min_{\boldsymbol{\theta} \in \boldsymbol{\Theta}} \mathbb{E}_{p(\tau)} \left[ -\mathbb{E}_{\pi_{\boldsymbol{\theta}'}, \mathcal{D}_\tau^Q} \left[ \sum_{t=0}^{H} \gamma^t r_t \right]; \boldsymbol{\theta}' = \boldsymbol{\theta} - \lambda \nabla_{\boldsymbol{\theta}} \left( -\mathbb{E}_{\pi_{\boldsymbol{\theta}}, \mathcal{D}_\tau^S} \left[ \sum_{t=0}^{H} \gamma^t r_t \right] \right) \right], \tag{13b}$$

where $\mathcal{D}_\tau$ denotes episodes collected from MDPs specified by $\tau$ under either the meta-policy or the fast-adapted policy. The term inside the brackets specifies the adaptation risk $\ell(\mathcal{D}_\tau^Q, \mathcal{D}_\tau^S; \boldsymbol{\theta})$, which represents either the negative cumulative returns or the negative final rewards. The expression $\boldsymbol{\theta} - \lambda \nabla_{\boldsymbol{\theta}} \ell(\mathcal{D}_\tau^S; \boldsymbol{\theta})$ indicates the gradient update for fast task adaptation, where $\lambda$ is the learning rate. Upon completion of meta-training, the resulting meta-policy $\boldsymbol{\theta}$ can be generalized across the task space.

**Robotics Domain Randomization.** Robotics domain randomization refers to a training paradigm in which an agent is trained across a collection of diverse environments to develop a generalizable policy. The diversity of these environments enhances the robustness of the resulting policy during deployment. Notably, this approach eliminates the need for few-shot episodes in unseen but similar environments. Mathematically, the optimization objective can be expressed as:

$$\max_{\boldsymbol{\theta} \in \boldsymbol{\Theta}} \mathcal{J}(\boldsymbol{\theta}) := \mathbb{E}_{\pi_{\boldsymbol{\theta}}} \mathbb{E}_{p(\tau)} \left[ \sum_{t=0}^{H} \gamma^t r_t \right], \tag{14}$$

where $p(\tau)$ denotes the distribution over MDPs specified by $\boldsymbol{\tau}$, and $\{r_t\}_{t=0}^{H}$ represents the stepwise rewards obtained from interacting with a specific MDP, with $H$ as the horizon.

After solving the optimization problem in Eq. (14), the resulting policy $\pi_{\boldsymbol{\theta}}$ serves as a zero-shot decision-maker in new environments. In this context, the adaptation risk can be expressed as $\ell(\mathcal{D}_\tau^Q, \mathcal{D}_\tau^S; \boldsymbol{\theta}) = -\sum_{t=0}^{H} \gamma^t r_t$. For physical domain randomization, policy optimization utilizes the TD3 algorithm (Fujimoto et al., 2018), an off-policy method known for its sample efficiency and stability. For visual domain randomization, we employ the PPO algorithm (Schulman et al., 2017), an on-policy method recommended by ManiSkill3 (Tao et al., 2024).

## A.2. Pseudo Algorithm in Meta-RL and DR

In this part, we show how PDTS is incorporated in DR and Meta-RL. These are Algorithm 2/3 and Algorithm 4/5.

---

**Algorithm 2** PDTS for Domain Randomization (Zero-Shot Scenarios)

---

**Input** : Task distribution $p(\tau)$; Task batch size $\mathcal{B}$; Learning rate $\lambda_1$.

**Output** : Adapted policy $\boldsymbol{\theta}$.

Set the initial iteration number $t = 1$;

Randomly initialize policy $\boldsymbol{\theta}$;

Randomly initialize risk learner $\{\boldsymbol{\psi}, \boldsymbol{\phi}\}$;

**while** *not converged* **do**

    Execute **Algorithm** 3 to access the task batch $\{\boldsymbol{\tau}_{t,i}\}_{i=1}^{\mathcal{B}}$;

    Sample trajectories for each task with policy $\boldsymbol{\theta}$ to induce $\{\mathcal{D}_{\boldsymbol{\tau}_{t,i}}^{Q}\}_{i=1}^{\mathcal{B}}$;

    // **Eval Adaptation Performance**

    Compute the task specific adaptation risk $\{\ell_{t,i} := -\mathbb{E}_{\pi_{\boldsymbol{\theta}}, \mathcal{D}_{\boldsymbol{\tau}_{t,i}}^{Q}}\left[\sum_{t=0}^{H} \gamma^t r_t\right]\}_{i=1}^{\mathcal{B}}$;

    Return $H_t = \{[\boldsymbol{\tau}_{t,i}, \ell_{t,i}]\}_{i=1}^{\mathcal{B}}$ as the Input to **Algorithm** 3;

    // **Update Policy**

    Perform batch gradient updates:

    $\boldsymbol{\theta}_{t+1} \leftarrow \boldsymbol{\theta}_t - \frac{\lambda_1}{\mathcal{B}} \sum_{i=1}^{\mathcal{B}} \nabla_{\boldsymbol{\theta}} \ell_{t,i}$;

    Update the iteration number: $t \leftarrow t + 1$;

**end**

---

**Algorithm 4** PDTS for Model Agnostic Meta Learning (Few-Shot Scenarios: Meta-RL, Sinusoid Regression)

---

**Input** : Task distribution $p(\tau)$; Task batch size $\mathcal{B}$; Learning rates: $\{\lambda_{1,1}, \lambda_{1,2}\}$.

**Output** : Meta-trained initialization $\boldsymbol{\theta}^{\text{meta}}$.

Set the initial iteration number $t = 1$;

Randomly initialize meta policy $\boldsymbol{\theta}^{\text{meta}}$;

Randomly initialize risk learner $\{\boldsymbol{\psi}, \boldsymbol{\phi}\}$;

**while** *not converged* **do**

    Execute **Algorithm** 5 to access the batch $\{\boldsymbol{\tau}_{t,i}\}_{i=1}^{\mathcal{B}}$;

    Sample trajectories for each task with policy $\boldsymbol{\theta}^{\text{meta}}$ to induce $\{\mathcal{D}_{\boldsymbol{\tau}_{t,i}}^{S}\}_{i=1}^{\mathcal{B}}$;

    // **Inner Loop to Fast Adapt**

    **for** $i = 1$ **to** $K$ **do**

        Compute the task-specific adaptation risk:

        $\ell(\mathcal{D}_{\boldsymbol{\tau}_{t,i}}^{S}; \boldsymbol{\theta}) = -\mathbb{E}_{\pi_{\boldsymbol{\theta}}, \mathcal{D}_{\boldsymbol{\tau}_{t,i}}^{S}}\left[\sum_{t=0}^{H} \gamma^t r_t\right]$;

        Perform gradient updates as fast adaptation:

        $\boldsymbol{\theta}_t^i \leftarrow \boldsymbol{\theta}_t^{\text{meta}} - \lambda_{1,1} \nabla_{\boldsymbol{\theta}} \ell(\mathcal{D}_{\boldsymbol{\tau}_i}^{S}; \boldsymbol{\theta})$;

        Sample trajectories $\mathcal{D}_{\boldsymbol{\tau}_{t,i}}^{Q}$ with policy $\boldsymbol{\theta}_t^i$ for task $\boldsymbol{\tau}_i$;

    **end**

    // **Outer Loop to Meta-train**

    Evaluate fast adaptation performance $\{\ell_{t,i} := -\mathbb{E}_{\pi_{\boldsymbol{\theta}_t^i}, \mathcal{D}_{\boldsymbol{\tau}_{t,i}}^{Q}}\left[\sum_{t=0}^{H} \gamma^t r_t\right]\}_{i=1}^{\mathcal{B}}$;

    Return $H_t = \{[\boldsymbol{\tau}_{t,i}, \ell_{t,i}]\}_{i=1}^{\mathcal{B}}$ as the Input to **Algorithm** 5;

    Perform meta initialization updates:

    $\boldsymbol{\theta}_{t+1}^{\text{meta}} \leftarrow \boldsymbol{\theta}_t^{\text{meta}} - \frac{\lambda_{1,2}}{\mathcal{B}} \sum_{i=1}^{\mathcal{B}} \nabla_{\boldsymbol{\theta}} \ell_{t,i}$;

    Update the iteration number: $t \leftarrow t + 1$;

**end**

---

**Algorithm 3** Posterior-Diversity Synergized Task Sampling

---

**Input** : Task distribution $p(\tau)$; Task batch size $\mathcal{B}$; Candidate batch size $\hat{\mathcal{B}}$; Latest updated $\{\boldsymbol{\psi}, \boldsymbol{\phi}\}$; Latest history $H_{t-1}$; Iteration number $K$; Learning rate $\lambda_2$.

**Output** : Selected task identifier batch $\{\boldsymbol{\tau}_{t,i}\}_{i=1}^{\mathcal{B}}$.

// **Optimize Risk Predictive Module**

**for** $i = 1$ **to** $K$ **do**

    Perform gradient updates given $H_{t-1}$:

    $\boldsymbol{\phi} \leftarrow \boldsymbol{\phi} + \lambda_2 \nabla_{\boldsymbol{\phi}} \mathcal{G}_{\text{ELBO}}(\boldsymbol{\psi}, \boldsymbol{\phi})$ in Eq. (5);

    $\boldsymbol{\psi} \leftarrow \boldsymbol{\psi} + \lambda_2 \nabla_{\boldsymbol{\psi}} \mathcal{G}_{\text{ELBO}}(\boldsymbol{\psi}, \boldsymbol{\phi})$ in Eq. (5);

**end**

// **Simulating After-Adaptation Results**

Randomly sample $\{\hat{\boldsymbol{\tau}}_{t,i}\}_{i=1}^{\hat{\mathcal{B}}}$ from $p(\tau)$;

// **Posterior Sampling Outcome**

Amortized evaluation $\{\hat{\ell}_{t,i}\}_{i=1}^{\hat{\mathcal{B}}}$ with one stochastic forward pass for all candidate tasks through executing Eq. (12)a-b;

// **Diversity-Guided Subset Search**

Run approximate algorithms to solve Eq. (12)c;

Return the screened subset $\{\boldsymbol{\tau}_{t,i}\}_{i=1}^{\mathcal{B}}$ for next iteration.

---

**Algorithm 5** Posterior-Diversity Synergized Task Sampling

---

**Input** : Task distribution $p(\tau)$; Task batch size $\mathcal{B}$; Candidate batch size $\hat{\mathcal{B}}$; Latest updated $\{\boldsymbol{\psi}, \boldsymbol{\phi}\}$; Latest history $H_{t-1}$; Iteration number $K$; Learning rate $\lambda_2$.

**Output** : Selected task identifier batch $\{\boldsymbol{\tau}_{t,i}\}_{i=1}^{\mathcal{B}}$.

// **Optimize Risk Predictive Module**

**for** $i = 1$ **to** $K$ **do**

    Perform gradient updates given $H_{t-1}$:

    $\boldsymbol{\phi} \leftarrow \boldsymbol{\phi} + \lambda_2 \nabla_{\boldsymbol{\phi}} \mathcal{G}_{\text{ELBO}}(\boldsymbol{\psi}, \boldsymbol{\phi})$ in Eq. (5);

    $\boldsymbol{\psi} \leftarrow \boldsymbol{\psi} + \lambda_2 \nabla_{\boldsymbol{\psi}} \mathcal{G}_{\text{ELBO}}(\boldsymbol{\psi}, \boldsymbol{\phi})$ in Eq. (5);

**end**

// **Simulating After-Adaptation Results**

Randomly sample $\{\hat{\boldsymbol{\tau}}_{t,i}\}_{i=1}^{\hat{\mathcal{B}}}$ from $p(\tau)$;

// **Posterior Sampling Outcome**

Amortized evaluation $\{\hat{\ell}_{t,i}\}_{i=1}^{\hat{\mathcal{B}}}$ with one stochastic forward pass for all candidate tasks through executing Eq. (12)a-b;

// **Diversity-Guided Subset Search**

Run approximate algorithms to solve Eq. (12)c;

Return the screened subset $\{\boldsymbol{\tau}_{t,i}\}_{i=1}^{\mathcal{B}}$ for next iteration.

---

## A.3. Related Work

**Risk-Averse Methods and Robustness Evaluation.** Real-world decision-making scenarios are risk-sensitive (Chow et al., 2021), and this makes robust optimization an indispensable component of policy optimization (Tamar et al., 2015; Chow et al., 2015; Chow, 2017; Chow et al., 2018; Pan et al., 2019; Rigter et al., 2021; Greenberg et al., 2022). Recent advances turn to some risk-averse measures to improve robustness in the task distribution. These are detailed in Section 2.1, which includes the DRM (Lv et al., 2024; Wang et al., 2024c), GDRM (Sagawa et al., 2019), and MPTS (Wang et al., 2025). The existing worst-case optimization in meta-learning is (Collins et al., 2020), which relies on special relaxation tricks; otherwise, the optimization can be unstable. Besides, other inspiring robust methods in task sampling (Qi et al., 2024) have not been applied in DR or Meta-RL. As MPTS is the latest SOTA RATS method, we mainly compared PDTS with it in experiments. RoML (Greenberg et al., 2024) also considers the CVaR optimization in Meta-RL; hence, we include some comparisons given the PEARL (Rakelly et al., 2019) backbone. In terms of evaluation, the subpopulation shift and out-of-distribution scenarios are commonly used in the field (Koh et al., 2021).

**Cross-Task Adaptation in Sequential Decision-Making.** This work focuses on the zero-shot and few-shot decision-making scenarios. In zero-shot decision-making, the primary technique is to randomize the environments and train the agent in the distribution over environments, which is called domain randomization (Tobin et al., 2017; Muratore et al., 2018; Mehta et al., 2020; Muratore et al., 2021; Tiboni et al., 2023). Meta-learning is a promising paradigm for achieving few-shot adaptation to unseen tasks, avoiding learning from scratch (Hospedales et al., 2021). The secret behind its few-shot adaptation capability is to leverage past experience and consolidate these as the prior for fast problem-solving. There are three primary types of policy adaptation methods that can be seamlessly incorporated into deep RL scenarios. (i) The optimization-based methods aim to seek a robust meta-initialization of the model that can be adapted to unseen scenarios through fine-tuning (Finn et al., 2017; 2018; Abbas et al., 2022; Rajeswaran et al., 2019; Yoon et al., 2018; Gupta et al., 2018; Qi et al., 2024). (ii) The context-based methods mostly adopt an encoder and decoder structure in policy networks (Xu et al., 2021; Wang & Van Hoof, 2022a;b; Rakelly et al., 2019; Zintgraf et al., 2019; Li et al., 2020). The encoder summarizes the support trajectories into a latent variable to guide the policy adaptation. (ii) The recurrent methods mainly employ a recurrent neural network to encode the sequential decision-making episodes as the adaptation prior (Duan et al., 2016; Ritter et al., 2018). All of these zero-shot or few-shot learning is performed in a task episodic way, which means that a batch of MDPs are resampled in each iteration to train the adaptive policy during DR and Meta-RL.

Curriculum learning is also a crucial topic related to adaptive decision-making. Dennis et al. (2020) develop unsupervised environment design (UED) as a novel paradigm for environment distribution generation and achieve SOTA zero-shot transfer. Jiang et al. (2021) cast prioritized level replay to enhance UED and formulate dual curriculum design for improving OOD and zero-shot performance. In (Koprulu et al., 2023), heavy-tailed distributions are incorporated into the automated curriculum, which leads to robustness improvement. Wang et al. (2024b) propose to generate task distributions for meta-RL through adversarial training normalizing flows, which provides a data-driven experimental design pipeline for increasing adaptation robustness. In contrast, our work emphasizes robust task adaptation under a fixed task distribution. Integrating the idea of surrogate evaluation from PDTS into curriculum design could be an interesting direction for future research.

## A.4. CVaR$_\alpha$ as Risk-Averse Metrics

**Definition A.2** (CVaR$_\alpha$). *With $\boldsymbol{\theta}$-parameterized model, e.g., a deep RL policy, we can induce a random variable $\ell_i := \ell(\mathcal{D}_{\tau_i}^Q, \mathcal{D}_{\tau_i}^S; \boldsymbol{\theta})$ from $p(\tau)$. Then, we can define the cumulative adaptation risk distribution and its quantile by $F(\ell)$ and $\ell^\alpha = \min_\ell\{\ell|F(\ell) \geq \alpha\}$. Following (Rockafellar et al., 2000), CVaR at $\alpha$-level robustness can be expressed as:*

$$\text{CVaR}_\alpha[\ell(\mathcal{T}; \boldsymbol{\theta})] = \int \ell dF^\alpha(\ell; \boldsymbol{\theta}), \tag{15}$$

*Accordingly, the normalized tail risk task distribution is*

$$F^\alpha(\ell; \boldsymbol{\theta}) = \begin{cases} 0, & l < \ell^\alpha \\ \frac{F(\ell; \boldsymbol{\theta}) - \alpha}{1 - \alpha}, & l \geq \ell^\alpha, \end{cases} \tag{16}$$

*with $p_\alpha(\tau; \boldsymbol{\theta})$ as its probability density function.*

**Assumption 1** (Lipschitz Continuity). The adaptation risk function $\ell(\cdot; \boldsymbol{\theta})$ is $\beta_\tau$-Lipschitz continuous w.r.t. $\boldsymbol{\theta}$ and $\beta_\theta$-

Lipschitz w.r.t. $\boldsymbol{\tau}$, i.e.,

$$|\ell(\mathcal{D}_\tau^Q, \mathcal{D}_\tau^S; \boldsymbol{\theta}) - \ell(\mathcal{D}_\tau^Q, \mathcal{D}_\tau^S; \boldsymbol{\theta}')| \leq \beta_\tau ||\boldsymbol{\theta} - \boldsymbol{\theta}'||$$
$$|\ell(\mathcal{D}_\tau^Q, \mathcal{D}_\tau^S; \boldsymbol{\theta}) - \ell(\mathcal{D}_{\tau'}^Q, \mathcal{D}_{\tau'}^S; \boldsymbol{\theta})| \leq \beta_\theta ||\boldsymbol{\tau} - \boldsymbol{\tau}'||, \tag{17}$$

where $\forall \{\boldsymbol{\theta}, \boldsymbol{\theta}'\} \in \boldsymbol{\Theta}$ and $\forall \{\boldsymbol{\tau}, \boldsymbol{\tau}'\} \in \mathcal{T}$.

**Assumption 2** (Bounded Functions). Given arbitrary $\boldsymbol{\theta} \in \boldsymbol{\Theta}$ and the task $\tau \in \mathcal{T}$, the adaptation risk function $\ell(\cdot; \boldsymbol{\theta})$ satisfies:

$$\max_{\tau \in \mathcal{T}} \ell(\mathcal{D}_\tau^Q, \mathcal{D}_\tau^S; \boldsymbol{\theta}) \leq \ell_{\max}. \tag{18}$$

$\text{CVaR}_\alpha$ is a risk-averse measure in computational finance and management science (Linsmeier & Pearson, 2000; Rockafellar et al., 2000), and we write it in the Definition A.2. As for Assumptions 1 and 2, these are commonly seen in analysis (Lv et al., 2024; Wang et al., 2024c) and are also necessary to MPTS and PDTS. These constitute the predictability of the adaptation risk value over iterations as the relative task difficulty remains invariant after the model's parameter is perturbed a bit.

### A.5. Details in Risk Predictive Modules

As introduced in Section 2.2, a key feature of RATS is its use of risk predictive models, e.g., $p(\ell|\boldsymbol{\tau}, H_{1:t}; \boldsymbol{\theta}_t)$, as surrogates for expensive evaluations. To the best of our knowledge, MPTS (Wang et al., 2025) is the first to develop such a model for robust optimization. As introduced in Section 2.2, MPTS leverages generative modeling for adaptation optimization and employs approximate posterior inference to forecast adaptation risk values after one-step optimization. Since small deviations in the machine learner's parameters do not alter the relative difficulty scores within the task batch, this provides a tractable approach to coarse-grained task difficulty evaluation. The probabilistic graphical model is provided in Fig. 7.

Empirically, a series of evaluation on DR and Meta-RL in (Wang et al., 2025) has validated the plausibility of such a schema. Particularly, it achieves high Pearson correlation coefficient (PCC) values between the risk learner's predicted adaptation risk values, $\{\bar{\ell}_{t+1,i} :\approx \mathbb{E}_{q_\phi(\boldsymbol{z}_t|H_t)}[p_\psi(\ell|\boldsymbol{\tau}_{t+1,i}, H_{1:t})]\}_{i=1}^{\mathcal{B}}$ and the corresponding exact adaptation risk values $\{\ell_{t+1,i}\}_{i=1}^{\mathcal{B}}$, indicating the risk predictive model's ability to score difficulty. The Pearson correlation coefficient value is computed as
$\rho_{\bar{\ell},\ell} := \frac{\sum_{i=1}^{\mathcal{B}}(\bar{\ell}_{t+1,i} - \text{Mean}[\{\bar{\ell}_{t+1,.}\}])(\ell_{t+1,i} - \text{Mean}[\{\ell_{t+1,.}\}])}{\sqrt{\sum_{i=1}^{\mathcal{B}}(\bar{\ell}_{t+1,i} - \text{Mean}[\{\bar{\ell}_{t+1,.}\}])^2}\sqrt{\sum_{i=1}^{\mathcal{B}}(\ell_{t+1,i} - \text{Mean}[\{\ell_{t+1,.}\}])^2}}$.

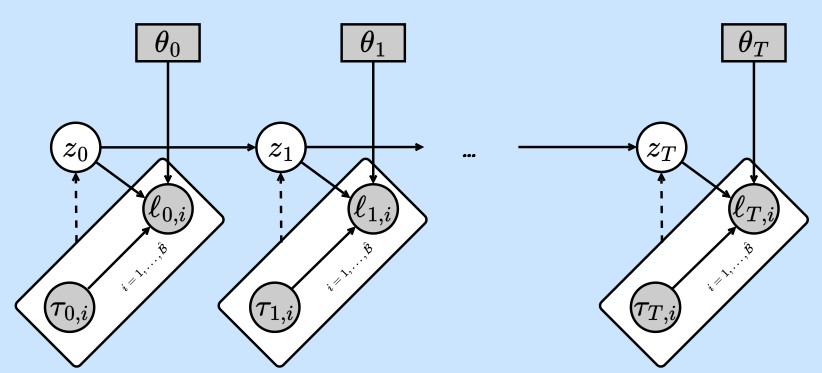

*Figure 7.* The probabilistic graphical model of the risk predictive model in (Wang et al., 2025), where gray units denote observed variables with the white as unobservable ones. The solid directed lines depict the generative model, and the dashed directed lines indicate the recognition model and approximate inference (Kingma & Welling, 2013).

**Formulation of ELBO & Stochastic Gradient Estimates** For the sake of easy and fast understanding of PDTS for the reviewer, we include the risk predictive model part as follows. These are modified from the MPTS team's open-sourced manuscript in (Wang et al., 2025). Here, the optimization objective of the risk predictive model is in Eq. (5) The risk predictive model uses a latent variable to summarize historical information and quantify uncertainty in predicting

task-specific adaptation risk. The following outlines the steps for deriving the evidence lower bound in optimization.

$$\mathcal{L}_{\text{ML}}(\boldsymbol{\psi}) := \ln p_{\boldsymbol{\psi}}(H_t|H_{1:t-1}) = \ln\Big[\int p_{\boldsymbol{\psi}}(H_t|\boldsymbol{z}_t)p(\boldsymbol{z}_t|H_{1:t-1})d\boldsymbol{z}\Big] \tag{19a}$$

$$= \ln\Big[\int q_{\boldsymbol{\phi}}(\boldsymbol{z}_t|H_t)\frac{p(\boldsymbol{z}_t|H_{1:t-1})}{q_{\boldsymbol{\phi}}(\boldsymbol{z}_t|H_t)}p_{\boldsymbol{\psi}}(H_t|\boldsymbol{z}_t)d\boldsymbol{z}_t\Big] \tag{19b}$$

$$\geq \mathbb{E}_{q_{\boldsymbol{\phi}}(\boldsymbol{z}_t|H_t)}\Big[\ln p_{\boldsymbol{\psi}}(H_t|\boldsymbol{z}_t)\Big] - D_{KL}\Big[q_{\boldsymbol{\phi}}(\boldsymbol{z}_t|H_t) \parallel p(\boldsymbol{z}_t|H_{1:t-1})\Big] := \mathcal{G}_{\text{ELBO}}(\boldsymbol{\psi},\boldsymbol{\phi}) \tag{19c}$$

Then, the ELBO is derived with the help of the reparameterization trick (Kingma & Welling, 2013). And Wang et al. (2025) further relax the ELBO to derive the $\beta$-VAE version, which corresponds to Eq. (5) in this work.

**Neural Architecture of the Risk Predictive Model.** For practicality, stability and fair comparison, we adopt the same risk predictive model design proposed in MPTS (Wang et al., 2025). As shown in Fig. 8, the risk learner follows an encoder-decoder structure. For consistency across benchmarks, we employ the same neural architecture similar with that of neural processes (Garnelo et al., 2018; Wang et al., 2023a) for all experiments. The encoder consists of an embedding network with four hidden layers, each containing 10 units and using Rectified Linear Unit (ReLU) activations. It encodes the batch $\{[\boldsymbol{\tau}_{t,i},\ell_{t,i}]\}_{i=1}^{\mathcal{B}}$ into $\boldsymbol{r}$ through mean pooling, subsequently mapping it to $[\boldsymbol{\mu},\boldsymbol{\sigma}]$. The latent variable $\boldsymbol{z}$ is sampled from a normal distribution defined by $[\boldsymbol{\mu},\boldsymbol{\sigma}]$. The decoder is a three-layer neural network with nonlinear activation functions, mapping $\{[\boldsymbol{\tau}_{t,i},\boldsymbol{z}]\}_{i=1}^{\mathcal{B}}$ to the predicted risk $\{\hat{\ell}_{t,i}\}_{i=1}^{\mathcal{B}}$. The optimization objective is given in Eq. (5). For further details on the implementation, please refer to our code repository.

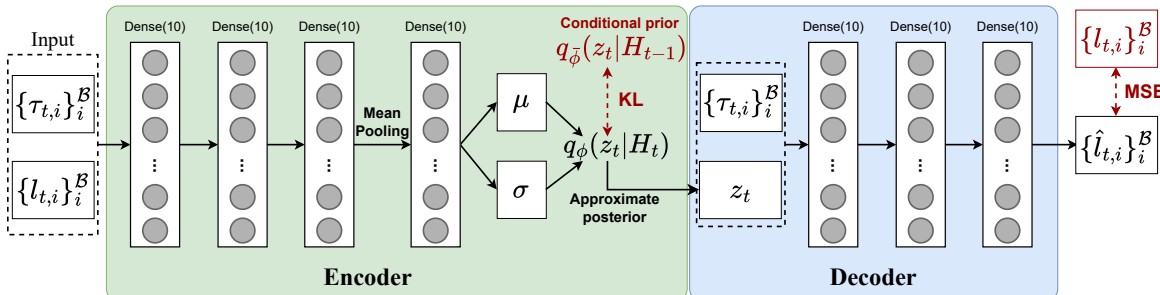

*Figure 8.* Illustration of the neural architecture of the risk predictive model in (Wang et al., 2025). The risk predictive model follows an encoder-decoder structure which encodes the batch $[\boldsymbol{\tau}_{t,i},\ell_{t,i}]$ into a latent variable $z$ and then decodes it into predicted adaptation risks $\hat{\ell}_{t,i}$. We reuse it in PDTS as a standard risk predictive model backbone.

# B. Theoretical Analysis and Proofs

## B.1. Preliminaries of MPTS

In MPTS (Wang et al., 2025), the streaming adaptation outcome in task episodic learning is characterized as:

$$\boldsymbol{\theta}_0 \xrightarrow{\text{eval}} \cdots \xrightarrow{\text{eval}} \{(\boldsymbol{\tau}_{t-1,i},\ell_{t-1,i})\}_{i=1}^{\mathcal{B}} \xrightarrow{\text{opt}} \boldsymbol{\theta}_t \xrightarrow{\text{eval}} \{(\boldsymbol{\tau}_{t,i},\ell_{t,i})\}_{i=1}^{\mathcal{B}} \xrightarrow{\text{opt}} \cdots \xrightarrow{\text{opt}} \boldsymbol{\theta}_T. \tag{20}$$

The most encouraging finding is that these cumulated task identifiers and adaptation risk values can be coupled to learn helpful adaptation prior and serve the evaluation of the task difficulty in a rough granularity. The resulting dataset is used to train the risk predictive model associated with the modified optimization objective as Eq. (5).

## B.2. One Secret MDP Steers Adaptation to Many MDPs

**The Operator for the State Transition in the Secret MDP.** Here, given a smooth and fixed machine learning optimizer, we can define the operator in zero-shot or few-shot adaptation optimization as:

$$\text{State Transition Operation after Adaptation} \quad \mathcal{F}: \boldsymbol{\theta} \times \mathcal{T}^{\mathcal{B}} \mapsto \boldsymbol{\theta}', \tag{21}$$

which corresponds to the deterministic transition function in the secret MDP $\mathcal{M}$. Here, take the DR case as an example, the above operator $\mathcal{F}$ results in the following probablly transited states:

$$\boldsymbol{\theta}_{t+1}^* = \arg\min_{\boldsymbol{\theta} \in \boldsymbol{\Theta}_{t+1}} \mathbb{E}_{p_\alpha(\tau;\boldsymbol{\theta})}\Big[\ell(\mathcal{D}_\tau^Q, \mathcal{D}_\tau^S; \boldsymbol{\theta})\Big],$$

$$\boldsymbol{\Theta}_{t+1} = \Big\{\boldsymbol{\theta}_t - \eta \frac{1}{\mathcal{B}} \nabla_{\boldsymbol{\theta}} \sum_{b=1}^{\mathcal{B}} \ell(\mathcal{D}_{\tau_b}^Q, \mathcal{D}_{\tau_b}^S; \boldsymbol{\theta}) | \tau_b \in \mathcal{T}_{t+1}^{\mathcal{B}}, |\mathcal{T}_{t+1}^{\mathcal{B}}| = \mathcal{B}, \mathcal{T}_{t+1}^{\mathcal{B}} \subseteq \mathcal{T}_{t+1}^{\hat{\mathcal{B}}} \Big\},$$

(22)

where $p_\alpha(\tau; \boldsymbol{\theta})$ corresponds to the probability density function of the $(1-\alpha)$-tailed tasks.

**Probabilistic Graphical Model of the Secret MDP.** Here, we can write the probabilistic form of the constructed MDP as:

$$p(\mathcal{T}_{1:T}^{\mathcal{B}}, R_{1:T}, \boldsymbol{\theta}_{0:T} | \Pi_{1:T}, H_{0:T-1}) = \underbrace{p(\boldsymbol{\theta}_0)}_{\text{Initial State}} \prod_{t=1}^{T} \underbrace{p(R_t | \boldsymbol{\theta}_{t-1}, \mathcal{T}_t^{\mathcal{B}})}_{\text{Step-Wise Reward}} \prod_{t=0}^{T-1} \underbrace{\pi_t(\mathcal{T}_{t+1}^{\mathcal{B}} | H_{0:t})}_{\text{Task Sampler}} \prod_{t=0}^{T-1} \underbrace{p(\boldsymbol{\theta}_{t+1} | \boldsymbol{\theta}_t, \mathcal{T}_{t+1}^{\mathcal{B}})}_{\text{State Transition}}. \quad (23)$$

where $\{\mathcal{T}_{0:T}^{\mathcal{B}}, r_{0:T}, \boldsymbol{\theta}_{0:T}\}$ records the trajectory information during policy search. The corresponding probabilistic graphical model is in Fig. 2.

### B.3. Proof of Proposition 3.2

**Proposition** 3.2 (MPTS as a UCB-guided Solution to i-MABs) *Executing MPTS pipeline in Eq. (6) is equivalent to approximately solving $\mathcal{M}$ with the i-MAB under the UCB principle.*

To demonstrate the claim in **Proposition** 3.2, we revisit fundamental concepts in $\text{CVaR}_\alpha$ optimization and break the problem into several steps. These include **# Step1** the dual form of $\text{CVaR}_\alpha$ with its Monte Carlo estimates, **Lemma B.1** to determine the optimal step-wise action, and **# Step2** nearly $\text{CVaR}_\alpha$ approximation/UCB-Guided Solution to i-MABs in MPTS.

**# Step1. The Greedy Action as the Subset with Top-$\mathcal{B}$ Adaptation Risk Values.**

**Unbiased Monte Carlo Estimate of $\text{CVaR}_\alpha$.** Here, we can rewrite the optimization objective of $\text{CVaR}_\alpha$ in the dual form:

$$\min_{\boldsymbol{\theta} \in \boldsymbol{\Theta}, \zeta \in \mathbb{R}} \text{CVaR}_\alpha(\boldsymbol{\theta}) := \zeta + \frac{1}{1-\alpha} \mathbb{E}_{p(\tau)}\Big[[\ell(\mathcal{D}_\tau^Q, \mathcal{D}_\tau^S; \boldsymbol{\theta}) - \zeta]^+\Big], \quad (24)$$

where the conditional function means $[\ell(\mathcal{D}_\tau^Q, \mathcal{D}_\tau^S; \boldsymbol{\theta}) - \zeta]^+ = \max\{\ell(\mathcal{D}_\tau^Q, \mathcal{D}_\tau^S; \boldsymbol{\theta}) - \zeta]^+, 0\}$. And its Monte Carlo sample average approximation corresponds to

$$\min_{\boldsymbol{\theta} \in \boldsymbol{\Theta}, \zeta \in \mathbb{R}} \text{CVaR}_\alpha(\boldsymbol{\theta}) := \zeta + \frac{1}{(1-\alpha)\mathcal{B}} \sum_{i=1}^{\mathcal{B}} [\ell(\mathcal{D}_\tau^Q, \mathcal{D}_\tau^S; \boldsymbol{\theta}) - \zeta]^+. \quad (25)$$

As the optimal auxiliary variable satisfies $\zeta = \text{VaR}_\alpha(\boldsymbol{\theta})$, the Top-$\mathcal{B}$ element in the set $\{\ell_i | \ell_i = \ell(\mathcal{D}_{\hat{\tau}_i}^Q, \mathcal{D}_{\hat{\tau}_i}^S; \boldsymbol{\theta})\}_{i=1}^{\hat{\mathcal{B}}}$ can be viewed as the unbiased Monte Carlo estimate w.r.t. Eq. (24) and the equivalent form of Eq. (25). In implementation, (Wang et al., 2024c; Lv et al., 2024) typically evaluate the machine learner's adaptation performance on candidate tasks, rank their values, and filter out $(1-\alpha)\hat{\mathcal{B}}$ to optimize.

**Lemma B.1** (Subset with Top-$\mathcal{B}$ Risk Values as the Optimal Arm). *Given the secret MDP $\mathcal{M}$ in Section 3.1 and its step-wsie reward $R(\boldsymbol{\theta}_t, \mathcal{T}_{t+1}^{\mathcal{B}}) := \text{CVaR}_\alpha(\boldsymbol{\theta}_t) - \text{CVaR}_\alpha(\boldsymbol{\theta}_{t+1})$, selecting the subset with average Top-$\mathcal{B}$ risk values inherently maximizes $R(\boldsymbol{\theta}_t, \mathcal{T}_{t+1}^{\mathcal{B}})$.*

*Proof.* Given the state $\boldsymbol{\theta}_t$ and the available action set $\mathbf{A}_t$ in $\mathcal{M}$, we need to show that selecting the $\mathcal{T}_{t+1}^{\mathcal{B}*} \in \mathbf{A}_t$ brings largest CVaR decrease as $\text{CVaR}_\alpha(\boldsymbol{\theta}_t) - \text{CVaR}_\alpha(\boldsymbol{\theta}_{t+1}^*)$.

Let $\mathcal{T}_{t+1}^{\mathcal{B}*}$ denote the subset with highest average Top-$\mathcal{B}$ risk values in the set $\{\ell_i | \ell_i = \ell(\mathcal{D}_{\hat{\tau}_i}^Q, \mathcal{D}_{\hat{\tau}_i}^S; \boldsymbol{\theta})\}_{i=1}^{\hat{\mathcal{B}}}$, which can be viewed as the Monte Carlo sample, i.e., unbiased estimate of $p_\alpha(\tau; \boldsymbol{\theta}_t)$. This is also in accordance with the dual form of

CVaR$_\alpha$ in Eq. (25). Meanwhile, we can define its transited state as $\boldsymbol{\theta}_{t+1}^*$. In a similar way, let some arbitrary feasible action, i.e., the subset be $\mathcal{T}_{t+1}^{\mathcal{B}'}$. we can define the corresponding transited state as $\boldsymbol{\theta}_{t+1}^*$. Inspired by the notation in Eq. (A.2), we can re-express the subset $\mathcal{T}_{t+1}^{\mathcal{B}'}$ as the Monte Carlo sample from another task distribution $q(\tau)$, which differs from $p_\alpha(\tau; \boldsymbol{\theta}_t)$.

Next, we can estimate the robustness improvement under different actions from the one-step Taylor expansion trick.

$$\text{Unbiased Objective } \mathcal{L}(\boldsymbol{\theta}) := \mathbb{E}_{p_\alpha(\tau; \boldsymbol{\theta}_t)}\left[\ell(\mathcal{D}_\tau^Q, \mathcal{D}_\tau^S; \boldsymbol{\theta})\right] \quad \text{Biased Objective } \hat{\mathcal{L}}(\boldsymbol{\theta}) := \mathbb{E}_{q(\tau)}\left[\ell(\mathcal{D}_\tau^Q, \mathcal{D}_\tau^S; \boldsymbol{\theta})\right] \tag{26a}$$

$$\boldsymbol{\theta}_{t+1}' = \boldsymbol{\theta}_t - \eta\nabla_{\boldsymbol{\theta}}\mathcal{L}(\boldsymbol{\theta}) \quad \hat{\boldsymbol{\theta}}_{t+1}' = \boldsymbol{\theta}_t - \eta\nabla_{\boldsymbol{\theta}}\hat{\mathcal{L}}(\boldsymbol{\theta}) \tag{26b}$$

$$\mathcal{L}(\boldsymbol{\theta}_{t+1}') = \mathcal{L}(\boldsymbol{\theta}_t) - \eta\nabla_{\boldsymbol{\theta}}\mathcal{L}(\boldsymbol{\theta})^T\nabla_{\boldsymbol{\theta}}\mathcal{L}(\boldsymbol{\theta}) + \mathcal{O}(||\boldsymbol{\theta}_{t+1}' - \boldsymbol{\theta}_t||_2^2) \Rightarrow \mathcal{L}(\boldsymbol{\theta}_t) - \mathcal{L}(\boldsymbol{\theta}_{t+1}') \approx \eta||\nabla_{\boldsymbol{\theta}}\mathcal{L}(\boldsymbol{\theta})||_2^2 \tag{26c}$$

$$\mathcal{L}(\boldsymbol{\theta}_t) - \mathcal{L}(\hat{\boldsymbol{\theta}}_{t+1}') \approx \eta\nabla_{\boldsymbol{\theta}}\mathcal{L}(\boldsymbol{\theta})^T\nabla_{\boldsymbol{\theta}}\hat{\mathcal{L}}(\boldsymbol{\theta}) = \eta||\nabla_{\boldsymbol{\theta}}\mathcal{L}(\boldsymbol{\theta})||_2^2\cos\alpha_q \leq \eta||\nabla_{\boldsymbol{\theta}}\mathcal{L}(\boldsymbol{\theta})||_2^2 = \mathcal{L}(\boldsymbol{\theta}_t) - \mathcal{L}(\boldsymbol{\theta}_{t+1}'), \tag{26d}$$

where we typically assume the norms of stochastic gradients for $\nabla_{\boldsymbol{\theta}}\mathcal{L}(\boldsymbol{\theta})^T$ and $\nabla_{\boldsymbol{\theta}}\hat{\mathcal{L}}(\boldsymbol{\theta})$ are the same and their angle is $\alpha_q$. As a result, we can see the optimal stochastic gradient should be the unbiased estimate of the Eq. (24), which corresponds to the Top-$\mathcal{B}$ tasks in the pseudo batch $\mathcal{T}_{t+1}^{\hat{\mathcal{B}}}$ with $\frac{\mathcal{B}}{\hat{\mathcal{B}}} = 1 - \alpha$. This completes the proof of **Lemma** B.1. $\quad\square$

Meanwhile, remember that in the task-selection MDP $\mathcal{M}$, the agent will never revisit the previous state $\boldsymbol{\theta}_t$ due to the nature of the stochastic gradient descent in the operator $\mathcal{F}$ in Eq. (21). Finally, we can claim that maximizing the state action value $Q(\boldsymbol{\theta}_t, \mathcal{T}_{t+1}^{\mathcal{B}})$ in the i-MAB actually corresponds to maximizing the step-wise reward due to the Bellman optimality in the main paper, and picking up the worst subset secretly maximizes $R(\boldsymbol{\theta}_t, \mathcal{T}_{t+1}^{\mathcal{B}})$ in a greedy way. This implies that $\pi_t^* = \arg\max_{\mathcal{T}_{t+1}^{\mathcal{B}} \subseteq \mathcal{T}_{t+1}^{\hat{\mathcal{B}}}} Q(\boldsymbol{\theta}_t, \mathcal{T}_{t+1}^{\mathcal{B}})$ and $Q(\boldsymbol{\theta}_t, \mathcal{T}_{t+1}^{\mathcal{B}}) \propto \frac{1}{\mathcal{B}}\sum_{i=1}^{\mathcal{B}}\ell_{t+1,i}$, where $\{\ell_{t+1,i}\}_{i=1}^{\mathcal{B}}$ denotes the evaluated adaptation performance of a feasible subset $\mathcal{T}_{t+1}^{\mathcal{B}}$ conditioned on $\boldsymbol{\theta}_t$. In the presence of RATS, the adaptation performance is evaluated by the risk predictive model $p(\ell|\tau, H_{1:t}; \boldsymbol{\theta}_t)$ in an amortized way, which suggests $Q(\boldsymbol{\theta}_t, \mathcal{T}_{t+1}^{\mathcal{B}}) \propto \frac{1}{\mathcal{B}}\sum_{i=1}^{\mathcal{B}}\hat{\ell}_{t+1,i}$ with $\hat{\ell}$ the predicted value.

# Step2. UCB-Guided Solution to i-MABs in MPTS.

**Assumption 3** (Randomized Adaptation Risk Value Function). *Given the secret MDP $\mathcal{M}$ in Section 3, we assume the distribution of the adaptation risk value follows an implicit Gaussian distribution, i.e., $p(\ell_{t+1,i}|\tau_i, H_{1:t}; \boldsymbol{\theta}_t) = \mathcal{N}(\mu_{t+1,i}, \sigma_{t+1}^2)$.*

Note that tasks with their identifiers are sampled in an *i.i.d.* way; this induces the conditional independence and the distribution of their summation as Eq. (27) with the help of Assumption 3.

$$p(\mathcal{L}_{t+1}^{\mathcal{B}}|\mathcal{T}_{t+1}^{\mathcal{B}}, H_{1:t}; \boldsymbol{\theta}_t) = \prod_{i=1}^{\mathcal{B}}p(\ell_{t+1,i}|\tau_i, H_{1:t}; \boldsymbol{\theta}_t) = \prod_{i=1}^{\mathcal{B}}\mathcal{N}(\mu_{t+1,i}, \sigma_{t+1,i}^2)$$

$$\implies p(\sum_{i=1}^{\mathcal{B}}\ell_{t+1,i}|\mathcal{T}_{t+1}^{\mathcal{B}}; \boldsymbol{\theta}_t) = \mathcal{N}(\sum_{i=1}^{\mathcal{B}}\mu_{t+1,i}, \sum_{i=1}^{\mathcal{B}}\sigma_{t+1,i}^2) := \mathcal{N}(\mu_{t+1}^{\mathcal{B}}, \sigma_{t+1}^{\mathcal{B}^2}) \tag{27}$$

At the same time, we find in MPTS, the multiple stochastic forward passes in Eq. (7) are performed to obtain the MC estimated distribution parameters $\{m(\ell_i) := \hat{\mu}_{t+1,i}, \sigma_i(\ell_i) := \hat{\sigma}_{t+1,i}\}$ for each task identifier $\tau_i$ in the batch $\mathcal{T}_{t+1}^{\mathcal{B}}$. This can be further associated with the factorization in the risk predictive model as Eq. (28):

$$p(\mathcal{L}_{t+1}^{\mathcal{B}}|\mathcal{T}_{t+1}^{\mathcal{B}}, H_{1:t}; \boldsymbol{\theta}_t) \approx \int p_\psi(\mathcal{L}_{t+1}^{\mathcal{B}}|\mathcal{T}_{t+1}^{\mathcal{B}}, \boldsymbol{z}_t)q_\phi(\boldsymbol{z}_t|H_{1:t})d\boldsymbol{z}_t \tag{28a}$$

$$= \int \prod_{i=1}^{\mathcal{B}}p(\ell_{t+1,i}|\tau_{t+1,i}, \boldsymbol{z}_t)q(\boldsymbol{z}_t|H_{1:t})d\boldsymbol{z}_t. \tag{28b}$$

The last step shows that the worst subset corresponds to the unbiased Monte Carlo estimate of CVaR$_\alpha$, and its sample average adaptation risk value can be treated as the proxy of the reward. Hence, picking up the subset with each element in the Top-$\mathcal{B}$ risk values is doing exploitation in robust fast adaptation. Rethinking MPTS's acquisition function in Eq. (7), we

can find the following inequality:

$$\sqrt{\sum_{i=1}^{\mathcal{B}} \sigma_{t+1,i}^2} \leq \sum_{i=1}^{\mathcal{B}} \sigma_{t+1,i} \quad \text{with} \quad \forall \sigma_{t+1,i} \in \mathbb{R}^+ \tag{29a}$$

$$\implies \gamma_1 \sqrt{\sum_{i=1}^{\mathcal{B}} \sigma_{t+1,i}^2} + \sum_{i=1}^{\mathcal{B}} \gamma_0 \mu_{t+1,i} \leq \sum_{i=1}^{\mathcal{B}} \gamma_1 \sigma_{t+1,i} + \gamma_0 \mu_{t+1,i} \tag{29b}$$

$$\implies \underbrace{\gamma_1 \sqrt{\sum_{i=1}^{\mathcal{B}} \sigma(\ell_i)^2} + \gamma_0 \sum_{i=1}^{\mathcal{B}} m(\ell_i)}_{\text{UCB}} \leq \underbrace{\sum_{i=1}^{\mathcal{B}} \gamma_1 \sigma(\ell_i) + \gamma_0 m(\ell_i)}_{\text{Approximate UCB}} := \mathcal{A}_{\mathrm{U}}(\mathcal{T}^{\mathcal{B}}), \tag{29c}$$

which means MPTS actually executes the approximate UCB to balance the exploitation (picking up the subset with the estimated worst performance) and the exploration of the task space (picking up the arm with the nearly highest epistemic uncertainty (Wang & Van Hoof, 2020) captured by the risk predictive model).

The above two steps complete the proof of **Proposition** 3.2.

### B.4. Proof of Proposition 3.3

**Proposition** 3.3 (Concentration Issue in Average Top-$\mathcal{B}$ Selection) *Let* $f(\boldsymbol{\tau}) : \mathbb{R}^d \to \mathbb{R}$ *be a unimodal and continuous function, where* $d \in \mathbb{N}^+$ *and* $\boldsymbol{\tau} \in \mathbb{R}^d$, *with a maximum value* $f(\boldsymbol{\tau}^*)$ *at* $\boldsymbol{\tau}^*$. *We uniformly sample a set of points* $\mathcal{T}^{\hat{\mathcal{B}}} = \{\boldsymbol{\tau}_i\}_{i=1}^{\hat{\mathcal{B}}}$, *where* $\boldsymbol{\tau}_i$ *are i.i.d. with a probability* $p_\epsilon$ *of falling within a* $\epsilon$-*neighborhood of* $\boldsymbol{\tau}^*$ *as* $|f(\boldsymbol{\tau}) - f(\boldsymbol{\tau}^*)| \leq \epsilon$. *Following MPTS, we select the Top-$\mathcal{B}$ samples with the largest function values, i.e.,*

$$\mathcal{T}^{\mathcal{B}} = \textit{Top-}\mathcal{B}(\mathcal{T}^{\hat{\mathcal{B}}}, f), \quad \hat{\mathcal{B}}, \mathcal{B} \in \mathbb{N}^+, \ \mathcal{B} \leq \hat{\mathcal{B}},$$

*For any* $\epsilon > 0$ *such that* $p_\epsilon < \frac{\hat{\mathcal{B}} - \mathcal{B} + 2}{\hat{\mathcal{B}} + 1}$, *the concentration probability*

$$\mathbb{P}\left(|f(\boldsymbol{\tau}) - f(\boldsymbol{\tau}^*)| \leq \epsilon \mid \forall \boldsymbol{\tau} \in \mathcal{T}^{\mathcal{B}}\right)$$

*increases with* $\hat{\mathcal{B}}$ *and converges to 1 with* $\hat{\mathcal{B}} \to \infty$.

*Proof.* We define $p_\epsilon$ as the probability that a random variable $\boldsymbol{\tau}$ uniformly sampled from the domain of definition falls within the neighborhood of the maximum value $\boldsymbol{\tau}^*$, i.e.,

$$p_\epsilon = \mathbb{P}\left(|f(\boldsymbol{\tau}) - f(\boldsymbol{\tau}^*)| \leq \epsilon \mid \boldsymbol{\tau} \sim \mathrm{Unif}(\cdot)\right).$$

Next, consider the probability that at least $\mathcal{B}$ random variables from the set $\mathcal{T}^{\hat{\mathcal{B}}} = \{\boldsymbol{\tau}_i\}_{i=1}^{\hat{\mathcal{B}}}$, where $\hat{\mathcal{B}}, \mathcal{B} \in \mathbb{N}^+$ and $\mathcal{B} \leq \hat{\mathcal{B}}$, are within the neighborhood of $\boldsymbol{\tau}^*$. Since the $\boldsymbol{\tau}_i$'s are i.i.d. and $\boldsymbol{\tau}_i \sim \mathrm{Unif}(\cdot)$, the probability is given by

$$P^{\hat{\mathcal{B}},\mathcal{B}} = 1 - \left[\sum_{i=1}^{\mathcal{B}} p_\epsilon^{\hat{\mathcal{B}}-i+1}(1 - p_\epsilon)^{i-1} \binom{\hat{\mathcal{B}}}{i-1}\right].$$

Since $f$ is a unimodal and continuous function, we can directly relate the concentration probability as

$$\mathbb{P}\left(|f(\boldsymbol{\tau}) - f(\boldsymbol{\tau}^*)| \leq \epsilon \mid \forall \boldsymbol{\tau} \in \mathcal{T}^{\mathcal{B}}\right) = P^{\hat{\mathcal{B}},\mathcal{B}}.$$

To establish the monotonicity of $P^{\hat{\mathcal{B}},\mathcal{B}}$ with respect to $\hat{\mathcal{B}}$, observe that the term $p_\epsilon^{\hat{\mathcal{B}}-i+1}(1 - p_\epsilon)^{i-1}\binom{\hat{\mathcal{B}}}{i-1}$ is monotonically decreasing in $\hat{\mathcal{B}}$ for fixed $i$, given that $p_\epsilon < \frac{\hat{\mathcal{B}}-i+2}{\hat{\mathcal{B}}+1}$. To see this, we compute the ratio of consecutive terms:

$$\frac{p_\epsilon^{\hat{\mathcal{B}}-i+1}(1-p_\epsilon)^{i-1}\binom{\hat{\mathcal{B}}}{i-1}}{p_\epsilon^{\hat{\mathcal{B}}-i+2}(1-p_\epsilon)^{i-1}\binom{\hat{\mathcal{B}}+1}{i-1}} = \frac{\hat{\mathcal{B}}-i+2}{p_\epsilon(\hat{\mathcal{B}}+1)} > 1.$$

Thus, the sequence $p_\epsilon^{\hat{\mathcal{B}}-i+1}(1-p_\epsilon)^{i-1}\binom{\hat{\mathcal{B}}}{i-1}$ decreases with $\hat{\mathcal{B}}$, and consequently, $P^{\hat{\mathcal{B}},\mathcal{B}}$ increases monotonically in $\hat{\mathcal{B}}$ when $\mathcal{B}$ is fixed, provided that $p_\epsilon < \frac{\hat{\mathcal{B}}-\mathcal{B}+2}{n+1}$.

Therefore, we conclude that for any $\epsilon > 0$ such that $p_\epsilon < \frac{\hat{\mathcal{B}}-\mathcal{B}+2}{\hat{\mathcal{B}}+1}$, the probability $\mathbb{P}\left(|f(\boldsymbol{\tau})-f(\boldsymbol{\tau}^*)| \leq \epsilon \mid \forall \boldsymbol{\tau} \in \mathcal{T}^{\mathcal{B}}\right)$ increases monotonically with $\hat{\mathcal{B}}$ for fixed $\mathcal{B}$.

$\square$

### B.5. Proof of Proposition 3.4

**Proposition** 3.4 (Nearly Worst-Case Optimization with PDTS) *When $\hat{\mathcal{B}}$ grows large enough, optimizing the subset from Eq. (11) achieves nearly worst-case optimization.*

*Proof.* As the exact size of the subset $\mathcal{B}$ is fixed in the optimization, the ratio $\frac{\mathcal{B}}{\hat{\mathcal{B}}}$ goes to nearly 0 with the increase of $\hat{\mathcal{B}}$ to a certain scale. As the consequence, the number of arms grows to $C_{\hat{\mathcal{B}}}^{\mathcal{B}}$ and the robustness concept is $\text{CVaR}_{1-\frac{\mathcal{B}}{\hat{\mathcal{B}}}}$. Since the involvement of the diversity regularization perturbs the worst arm selection, this induces the nearly worst-case optimization in PDTS. $\square$

## C. Experimental Setups & Implementation Details

### C.1. Risk-Averse Baseline Details

These baselines are SOTA methods published in NeurIPS/ICLR conferences and the latest open-sourced version (Wang et al., 2024c; Lv et al., 2024; Sagawa et al., 2019; Wang et al., 2025; Greenberg et al., 2024).

**GDRM (Sagawa et al., 2019; Setlur et al., 2024).** As briefly introduced in the main paper, GDRM can be viewed as a min-max optimization problem. The core concept of enhancing the machine learner's robustness involves reallocating more probability mass to the worst-case scenarios in a weighted manner. In each iteration with the optimal $p_{\hat{g}}(\tau)$, the optimization problem simplifies to:

$$\min_{\boldsymbol{\theta} \in \boldsymbol{\Theta}} \mathbb{E}_{p_{\hat{g}}(\tau)}\left[\ell(\mathcal{D}_\tau^Q, \mathcal{D}_\tau^S; \boldsymbol{\theta})\right] = \mathbb{E}_{p(\tau)}\left[\frac{p_{\hat{g}}(\tau)}{p(\tau)}\ell(\mathcal{D}_\tau^Q, \mathcal{D}_\tau^S; \boldsymbol{\theta})\right], \tag{30}$$

where we use $\omega(\tau) = \frac{p_{\hat{g}}(\tau)}{p(\tau)}$ to denote the weight.

In general, for a fixed number of tasks, GDRM organizes tasks heuristically or dynamically into clusters, followed by a reweighting mechanism based on assessed risks. However, in task-episodic learning, no task grouping is performed because the task batch is reset after each iteration (Wang et al., 2024c; Lv et al., 2024). Task-specific weights are calculated as $\omega(\tau_i) = \frac{\exp(\eta\ell(\mathcal{D}_{\tau_i}^Q, \mathcal{D}_{\tau_i}^S; \boldsymbol{\theta}))}{\sum_{b=1}^{\mathcal{B}}\exp(\eta\ell(\mathcal{D}_{\tau_b}^Q, \mathcal{D}_{\tau_b}^S; \boldsymbol{\theta}))}$, where $\eta$ denotes the temperature parameter, and $\{\tau_b\}_{b=1}^{\mathcal{B}}$ represents the identifiers of the task batch. Additional implementation details are available at `https://github.com/kohpangwei/group_DRO`.

**DRM (Wang et al., 2024c; Lv et al., 2024).** As outlined in the main paper, we adopt the standard approach in DRM, where the optimization objective is expressed as $\mathbb{E}_{p_\alpha(\tau;\boldsymbol{\theta})}\left[\ell(\mathcal{D}_\tau^Q, \mathcal{D}_\tau^S; \boldsymbol{\theta})\right]$. A widely used practical strategy involves evaluating the performance of the machine learner, ranking task-specific adaptation risks, and optimizing over the worst $(1-\alpha)$ proportion of tasks.

Following the setup in Wang et al. (2024c), we select the Top-$\mathcal{B}$ tasks during optimization, which corresponds to evaluating a task batch of size $\frac{\mathcal{B}}{1-\alpha}$. To ensure a fair comparison with PDTS while preserving computational efficiency, we employ the same Monte Carlo estimator for the risk quantile as in Wang et al. (2024c). For all benchmarks, we set the actual task batch size to $\hat{\mathcal{B}} = 2\mathcal{B}$, discarding the easiest half before optimizing the machine learner.

**MPTS (Wang et al., 2025).** We have thoroughly introduced the MPTS pipeline in Section 2.2 and provided details of the risk learner in Section A.5. For additional configurations, we adopt those from the official repository at `https://github.com/thu-rllab/MPTS`, including the heuristic random mixture strategy and the specific values of $\hat{\mathcal{B}}$.

*Table 3.* **Details of Task Identifiers and Algorithm Backbones Across Benchmarks.** Here, we provide detailed information about the task identifiers used to induce task distributions and the algorithm backbones, including MAML (Finn et al., 2017), TD3 (Fujimoto et al., 2018), and PPO (Schulman et al., 2017), employed in various benchmarks.

| Benchmarks | Identifier Meaning | Identifier Range | Backbone |
|---|---|---|---|
| `K-shot` sinusoid regression | amplitude and phase $(a, b)$ | $[0.1, 5.0] \times [0, \pi]$ | MAML |
| Meta-RL: ReacherPos
Meta-RL: Walker2dVel
Meta-RL: Walker2dMassVel
Meta-RL: HalfCheetahMassVel | goal location $(x_1, x_2)$
velocity $v$
mass and velocity $(m, v)$
mass and velocity $(m, v)$ | $[-0.2, 0.2] \times [-0.2, 0.2]$
$[0, 2.0]$
$[0.75, 1.25] \times [0, 2.0]$
$[0.75, 1.25] \times [0, 2.0]$ | MAML |
| DR: Pusher
DR: LunarLander
DR: ErgoReacher | puck friction loss $f$ and puck joint damping $d$
main engine strength $s$
joint damping $d$ and max torque $t$ ($\times 4$ joints) | $[0.004, 0.01] \times [0.01, 0.025]$
$[4, 20]$
$[0.1, 2.0] \times [2, 20]$ | TD3 |
| VisualDR: LiftPegUpright_Light
VisualDR: AnymalCReach_Goal | ambient light $l$ (x3 dimensions)
goal location $(x_1, x_2)$ | $[-1.0, 2.0]$
$[0.0, 1.0] \times [0.0, 1.0]$ | PPO |

## C.2. Sinusoid Regression

Following standard setups in prior works (Finn et al., 2017; Wang et al., 2025), we define the sinusoid regression problem as a toy example of supervised meta-learning. The goal of this problem is to predict the wave function $y = a\sin(x - b)$, where the amplitude $a$ and phase $b$ are sampled as task-specific parameters, i.e., task identifier $\boldsymbol{\tau} = (a, b)$, using a few-shot support dataset. We adopt the 10-`shot` sinusoid regression setting, where 10 data points are uniformly sampled from the interval $[-5.0, 5.0]$ to form the support dataset for each task.

**Implementation Details.** The machine learner is a neural network with 2 hidden layers, each of size 40, using the Rectified Linear Unit (ReLU) as the nonlinear activation function. The task batch size is set to 16 for ERM and GDRM, while a batch size of 32 is used as the default for DRM. The temperature parameter in GDRM is $\eta = 0.001$, and the learning rates for both the inner and outer loops are fixed at $0.001$. The task identifier has a dimensionality of 2. For MPTS and PDTS, the identifier batch size during training is set to 32 ($2\times$) and 512 ($32\times$), respectively. We use the Adam optimizer with a learning rate of $5 \times 10^{-4}$ to update the risk learner over 15,000 steps. The label for the risk learner is the average MSE loss value for each task. For sinusoid regression and meta-reinforcement learning, we use the standard repository provided by MAML (Finn et al., 2017). Regarding validation during training, we use a separate uniform task sampler with a fixed random seed to select 1,000 tasks for validating the training checkpoints of various methods.

## C.3. Meta Reinforcement Learning

We adopt four Meta-RL scenarios—ReacherPos, Walker2dVel, Walker2dMassVel, and HalfCheetahMassVel—from MPTS (Wang et al., 2025), which represent distinct MDP distributions based on the Mujoco physics engine (Todorov et al., 2012). These scenarios involve three types of robots (HalfCheetah, Walker2d, and Reacher) and three randomized meta-learning objectives: velocity, goal location, and body mass.

- The objective in velocity-based scenarios, e.g., Walker2dVel, is to train the robot to achieve a target velocity, with the reward function defined as the negative absolute difference between the robot's current velocity and the target velocity. This reward is augmented by a control penalty and an alive bonus to facilitate learning. As the task distribution is specified by a uniform distribution over the target velocity, $\boldsymbol{\tau} = v$ can be viewed as the task identifier.

- The body mass and velocity scenarios, e.g., Walker2dMassVel and HalfCheetahMassVel, share the same objective as the velocity scenarios but additionally feature varying robot masses. As the task distribution is specified by a uniform distribution over the body mass and target velocity, $\boldsymbol{\tau} = (m, v)$ can be viewed as the task identifier.

- The goal location scenario, e.g., ReacherPos, requires moving a two-jointed robot arm's end effector close to a target position. Its reward function is defined as the negative $L_1$ distance between the end effector's position and the target,

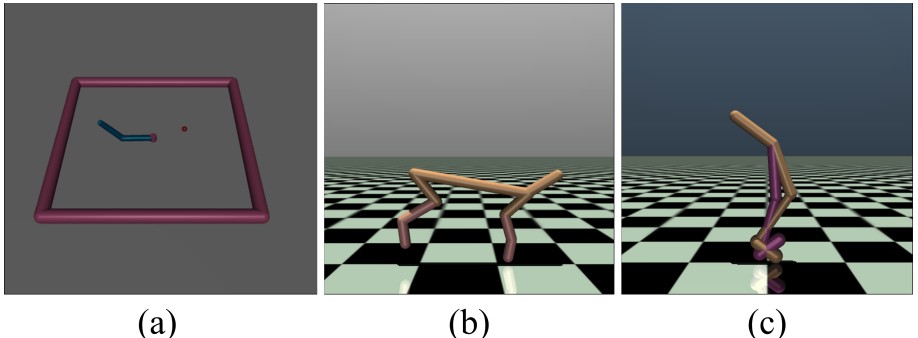

*Figure 9.* Illustrations of three types of robots in Meta-RL based on the Mujoco. (a) Reacher, (b) HalfCheetah, and (c) Walker2d.

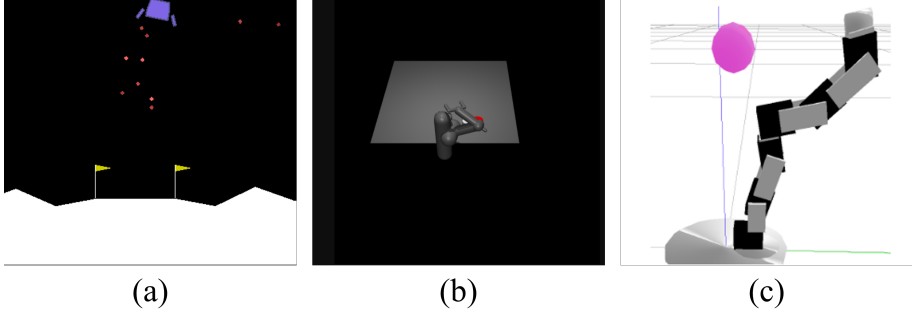

*Figure 10.* Illustrations of three physical robotics domain randomization scenarios: (a) LunarLander, (b) Pusher, and (c) ErgoReacher. The illustration of ErgoReacher is adapted from (Mehta et al., 2020).

supplemented by a control cost to encourage robustness. As the task distribution is specified by a uniform distribution over the goal location, $\boldsymbol{\tau} = (x_1, x_2)$ can be viewed as the task identifier.

**Implementation Details.** The machine learner is implemented as a neural network with 2 hidden layers, each consisting of 64 units, and utilizes ReLU activations for nonlinearity. For ERM and GDRM, the default task batch size is set to 20, while DRM uses a batch size of 40. The temperature parameter for GDRM is configured as 0.001. Both the inner and outer loop learning rates are fixed at 0.1. Besides, the identifier batch size during training is 30 ($1.5\times$) for MPTS and 1280 ($64\times$) for PDTS. The risk learner is updated using the Adam optimizer with a learning rate of $5 \times 10^{-3}$. The label for the risk learner is the negative average reward value at the final step for each task. For validation during training, we uniformly sample 40 tasks from the task space at fixed intervals to validate the training checkpoints of different methods. For meta-testing after training, we uniformly sample 100 tasks from the task space to test the trained models of different methods.

### C.4. Physical Robotics Domain Randomization

As shown in Fig. 10, we adopt three scenarios for robotics domain randomization from Mehta et al. (2020): Pusher, LunarLander, and ErgoReacher. The task distribution is defined as in (Wang et al., 2025). Specifically:

- LunarLander is a 2-degree-of-freedom (DoF) environment where the agent must softly land a spacecraft, implemented in Box2D (Catto, 2007). Its reward function provides positive rewards for successful landings, negative rewards for crashes, and penalties for fuel consumption and deviations from the landing pad, thereby promoting efficient and controlled landings. The task distribution is specified by a uniform distribution over the main engine strength $s$, which can be viewed as the task identifier $\boldsymbol{\tau} = s$.

- Pusher is a 3-DoF robotic arm control environment based on MuJoCo (Todorov et al., 2012), where the agent pushes a puck to a target. The reward function penalizes the $L_2$ distance between the puck and the target, augmented by a control penalty. The task distribution is specified by a uniform distribution over the puck friction loss $f$ and puck joint damping $d$, which can be viewed as the task identifier $\boldsymbol{\tau} = (f, d)$.

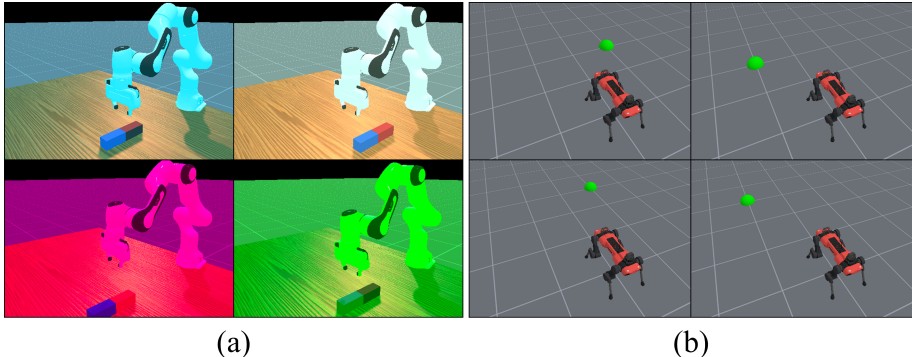

(a)             (b)

*Figure 11.* Illustrations of two visual robotics domain randomization scenarios: controlling (a) a table-top robotic arm and (b) a quadruped robot, operating under randomized lighting conditions and varying goal locations, respectively.

- ErgoReacher involves a 4-DoF robotic arm implemented in the Bullet Physics Engine (Coumans, 2015), tasked with reaching a goal using its end effector. Its reward function penalizes the distance between the end effector and the target, combined with control penalties. The task distribution is specified by a uniform distribution over the joint damping $d$ and max torque $t$ across 4 joints, which can be viewed as the task identifier $\tau = (d_1, d_2, d_3, d_4, t_1, t_2, t_3, t_4)$.

Details of the randomized task identifier range are presented in Table 3.

**Implementation Details.** The machine learner is a neural network with two hidden layers, each consisting of 10 units, and uses ReLU activation functions. For ERM and GDRM, the task batch size is set to 10, while DRM uses a batch size of 20. GDRM employs a temperature parameter of 0.01. We adopt TD3 algorithm (Fujimoto et al., 2018) as the algorithm backbone. The actor and critic learning rates are both set to $3 \times 10^{-4}$. For MPTS, the identifier batch size during training is 25 ($2.5\times$) for LunarLander, 50 ($5\times$), and 250 ($25\times$) for ErgoReacher. In contrast, for PDTS, the identifier batch size during training is 640 ($64\times$) for all environments, with no additional requirements for fine-tuning. The risk learner is updated using the Adam optimizer with a learning rate of 0.005. The label for the risk learner is the negative average return for each task. For validation during training, we uniformly sample 100 tasks from the task space to validate the training checkpoints of different methods.

### C.5. Visual Robotics Domain Randomization

Visual-based robotics control is common and crucial in real-world scenarios. As illustrated in Fig. 11, based on the latest robotics simulator, ManiSkill3 (Tao et al., 2024), we design two scenarios for visual robotics domain randomization: LiftPegUpright_Light and AnymalCReach_Goal. These scenarios involve controlling a tabletop two-finger gripper arm robot and a quadruped robot, respectively, under randomized lighting conditions and goal locations.

Specifically:

- LiftPegUpright_Light is derived from the LiftPegUpright-v1 scenario in ManiSkill3, where the objective is to move a peg lying on the table to an upright position. To emulate the complex lighting conditions found in real-world environments, we modify this scenario to make the ambient lighting controllable. An illustration of this setup is shown in Fig. 6. The task distribution is specified by a uniform distribution over the configurations of ambient light, which can be viewed as the task identifier $\tau = (l_1, l_2, l_3)$.

- AnymalCReach_Goal is adapted from the AnymalC-Reach-v1 scenario in ManiSkill3. The task is to control the AnymalC robot to reach a target location in front of it. Inspired by point-robot navigation scenarios in prior works(Finn et al., 2017), we randomize the goal location, which remains visible to the robot. The task distribution is specified by a uniform distribution over the goal location, which can be viewed as the task identifier $\tau = (x_1, x_2)$.

The detailed randomization configurations are provided in Table 3.

**Implementation Details.** We use the PPO algorithm (Schulman et al., 2017), along with its hyperparameters and network architecture, as provided in the official ManiSkill3 codebase. To accommodate GPU memory limitations, we set the

task batch size to 16 for ERM and GDRM and 32 for DRM, with 500 steps per iteration on LiftPegUpright_Light. For AnymalCReach_Goal, the task batch size is 128 for ERM and GDRM and 256 for DRM, with 200 steps per iteration. For MPTS, the identifier batch size during training is 32 for LiftPegUpright_Light and 256 for AnymalCReach_Goal (2.5×). For PDTS, the identifier batch size during training is $64\times$ the default for all environments. The risk learner is updated using the Adam optimizer with a learning rate of 0.005. The label for the risk learner is the negative average reward for each task. For validation during training, we uniformly sample 64 tasks from the task space to validate the training checkpoints of different methods.

## D. Additional Experiment Results

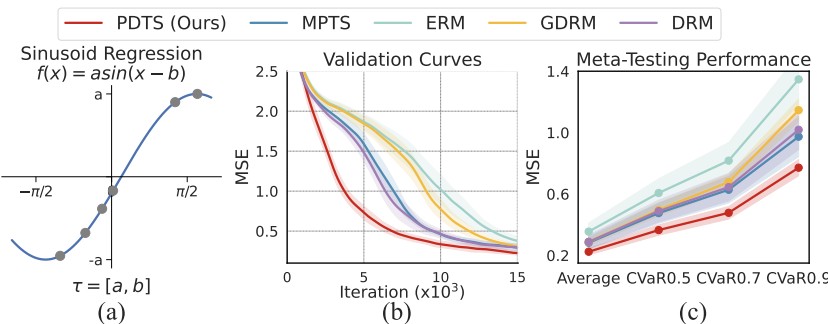

*Figure 12.* **Few-Shot Sinusoid Regression Results**. (a) Illustration of the sinusoid regression problem, where the task identifier $\tau$ consists of the amplitude and phase $[a, b]$. (b) Curves of averaged MSEs on the validation task set for all methods during meta-training. (c) MSE values for all methods at various $\alpha$ levels of $\text{CVaR}_\alpha$ during meta-testing.

### D.1. Beyond Decision-Making: Robust Supervised Meta-Learning

This work primarily focuses on risk-averse sequential decision-making. However, PDTS is readily extendable to other risk-averse scenarios, such as supervised meta-learning. As shown in Fig. 12, we evaluate PDTS on sinusoid regression, a commonly-used toy example introduced in Finn et al. (2017), which involves adapting quickly to new functions using only 10 samples. Consistent with the results observed in decision-making, PDTS achieves superior performance compared to all baselines, demonstrating faster average performance and more robust adaptation. Moreover, as demonstrated in MPTS (Wang et al., 2025), the RATS paradigm has broad applicability, including image classification (Gondal et al., 2024). It is believed to be promising in other interesting areas, such as LLM-guided decision-making (Ma et al., 2024; Wang et al., 2024a; Qu et al., 2024; 2025), multi-agent systems (Shao et al., 2023b;a; Qu et al., 2023), and data sampling in offline reinforcement learning (Levine et al., 2020; Zhang et al., 2023; Mao et al., 2023a;b; Hong et al., 2023; Mao et al., 2024a;b).

### D.2. Ablation Studies and Additional Analysis

In this section, we perform additional experiments to carry out ablation studies on the hyperparameters and components of PDTS, demonstrate the presence of the concentration issue in MPTS, and validate the effectiveness of PDTS in addressing it. For computational efficiency, we use sinusoid regression as the testbed.

**Ablation Study on Diversity Regularization Weight $\gamma$.**  As shown in Fig. 13(a), we evaluate the effect of varying values of the diversity regularization weight $\gamma$. It is evident that diversity regularization plays a crucial role in PDTS, as its absence ($\gamma = 0$) leads to a dramatic performance drop. PDTS demonstrates robustness to the choice of $\gamma$ within a certain range (e.g., $[1, 2]$). However, excessively large values of $\gamma$ degrade performance, emphasizing the importance of balancing diversity and robust optimization. In most cases, setting $\gamma$ to 1 or a nearby value secures superior enough performance. For scenarios with an extremely low-dimensional task identifier, increasing $\gamma$ appropriately may improve performance.

**Ablation Study on Key Components: Posterior Sampling and Diversity Regularization.**  In simple terms, PDTS can be viewed as the combination of MPTS and diversity regularization, with UCB replaced by posterior sampling. We conduct an ablation study to highlight the significance of each component. As shown in Fig. 13(b), we replace the UCB in MPTS with posterior sampling (MPTS+P) and incorporate diversity regularization into MPTS (MPTS+D). The results demonstrate that each component contributes significantly to the superiority of PDTS.

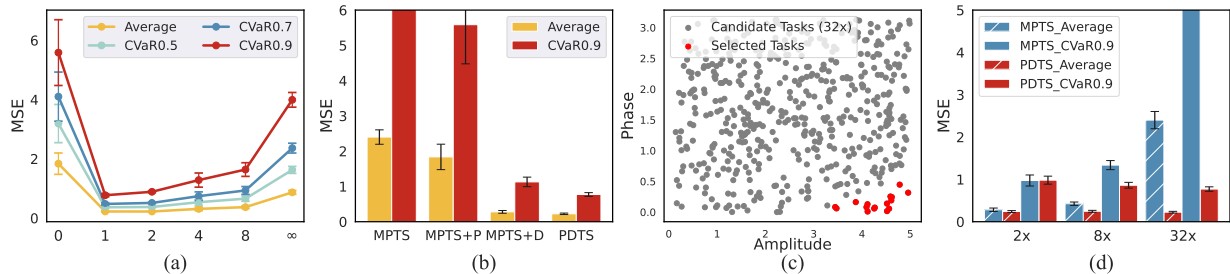

*Figure 13.* (a) Meta-testing results trained with different hyperparameter, $\gamma$. (b) Ablation studies on key components, including posterior sampling (P) and diversity regularization (D). (c) Visualization of the distribution of candidate tasks (gray points) and selected tasks (red points) in MPTS, highlighting the concentration issue. (d) Comparison of the average and $CVaR_{0.9}$ meta-testing performance of PDTS and MPTS with increasing pseudo-batch sizes.

**The Presence of the Concentration Issue in MPTS.** As introduced in Sec. 3.2, we theoretically prove the presence of a concentration issue in MPTS. To empirically demonstrate the concentration issue and its impact, we evaluate MPTS on sinusoid regression. Fig. 13(c) shows that, as the candidate batch size increases, MPTS tends to select tasks concentrated within a small region—a phenomenon referred to as the concentration issue in the main paper. As illustrated in Fig. 13(d), this concentration issue leads to catastrophic performance degradation in MPTS as the candidate batch size increases.

**PDTS Addresses the Concentration Issue and Benefits from Improved Coverage.** In contrast to MPTS, Fig. 13(e) demonstrates that by incorporating diversity regularization, our method, PDTS, avoids the concentration issue and does not experience performance collapse as the candidate batch size increases. More impressively, the performance of PDTS improves with increasing candidate batch size, demonstrating the benefits of encouraging broader coverage of the task space during subset selection as proposed by PDTS.

**Ablation Study on the Risk Predictive Model.** We conducted an ablation study to analyze the impact of the risk predictive model. We designed two variants, PDTS-Deep and PDTS-Shallow, by increasing and decreasing the number of encoder-decoder layers, respectively. Additionally, we replaced the encoder-decoder structure with an MLP to create PDTS-MLP. Results on Walker2dVel are summarized in Table 4. From these results, we observe: (1) All PDTS variants achieve better task robust adaptation, confirming the effectiveness of PDTS and its generality across different risk predictive models. (2) The encoder-decoder architecture generally outperforms MLP-based models, supporting the rationale behind this design. (3) Deeper networks may introduce a performance-robustness trade-off in the current setting, which we plan to further investigate in more complex scenarios. (4) Weaker risk prediction models degrade overall performance, due to poorer difficult MDP identification.

*Table 4.* Comparison of methods with different risk predictive models on Walker2dVel.

| Methods | $CVaR_{0.9}$ | $CVaR_{0.7}$ | $CVaR_{0.5}$ | Average |
|---------|---------|---------|---------|---------|
| ERM | -69.77±7.62 | -31.73±7.82 | -3.78±6.66 | 38.88±4.73 |
| PDTS | **-22.42±3.13** | **2.86±3.04** | 16.57±2.93 | 40.40±3.07 |
| PDTS-Deep | -30.51±7.11 | 0.55±5.69 | **17.99±4.8** | **44.42±3.69** |
| PDTS-Shallow | -41.24±4.74 | -11.34±4.5 | 3.92±4.33 | 33.64±4.01 |
| PDTS-MLP | -41.07±4.88 | -12.04±5.06 | 3.38±4.94 | 32.96±4.58 |

# E. Other Discussions

**Relation with Traditional Active Learning.** Traditional active learning (Ren et al., 2021) aims at reducing the sampling redundancies during optimization and exploiting historical optimization information to improve learning efficiency, such as annotations and computations. The active query strategies (Zhu et al., 2003; Gal et al., 2017; Kirsch et al., 2019; Wu et al., 2022; Mukhoti et al., 2023) also rely on some predictive models and utilize principles like uncertainty, diversity, etc. RATS, such as MPTS (Wang et al., 2025) and PDTS in this work, stresses the importance of robustness during active sampling. Hence, the predictive model requires scoring the task difficulties without exact evaluation.

**When MPTS Meets Diversity Regularization.** The diagnosis of the concentration issue in Sec. 3.2 identifies the diversity regularization as a plausible solution to encourage the exploration of the task space and bring more worst-case robust solutions as more arms are constructed by increasing $\mathcal{B}$. Actually, we also examine this part and include the ablation studies on the sinusoid regression in Fig. 13(b). For other evaluations in the main paper, we assume that MPTS's group (Wang et al., 2025) adopts the optimal hyper-parameter configurations. Hence, their setup is adopted to produce MPTS results. Meanwhile, the posterior sampling's advantage lies in (i) no extra hyperparameter adjustment, unlike UCB used in MPTS, and (ii) stochastic optimism when the uncertainty is difficult to estimate.

**PDTS is Agnostic to the Risk Predictive Model.** As RATS in this work is a rarely investigated concept in the field, limited methods have been developed to score task difficulties, particularly MDPs' difficulties under a policy. The risk predictive model in MPTS has approximately achieved the purpose. Hence, we reuse their module as the backbone. In reality, our theoretical analysis and PDTS will be compatible with other risk-predictive models in the future.

## F. Computational Platform & Software

This research project conducts experiments using NVIDIA 3090 GPUs in computation, and Pytorch works as the deep learning toolkit in implementation. The software requirement list can be found in the open-source code from the project website. Please refer to technical blog from our team website at `https://www.thuidm.com/`.

