# OpenReview forum: "Fast and Robust: Task Sampling with Posterior and Diversity Synergies for Adaptive Decision-Makers in Randomized Environments"
_ICML.cc/2025/Conference — ICML 2025 poster_

### Official Review · Reviewer_beLW · 2025-03-02

**Overall Recommendation:** 5

**Summary:**

The authors study the problem of robust reinforcement learning through the lens of meta-RL in which the trained (meta) agent receives a few task-specific samples (can be zero) using which it adapts to a new task. The objective is to maximize the expected performance conditioned on the sampled task being in a specific bottom quantile w.r.t. performance (CVaR). The proposed approach is based on the RATS framework and an existing approach called MPTS which continuously learns a model (along with the meta-agent) that can be used to predict the expected reward (or loss) of the agent on a new task given historical information about the samples used to train the agent. This predictive model is then used to select new tasks for the next iteration of the meta-RL algorithm. The main contributions of this paper are (i) formulation of the task selection problem (which tasks to use in each iteration of the meta-RL algorithm) as a higher-level reinforcement learning problem in which the actions are subsets of tasks, (ii) connecting the task-selection algorithm of MPTS to UCB, and (iii) proposing a new task-selection procedure that includes diversity of selected tasks in its objective.

**Claims And Evidence:**

Yes. The theoretical claims seem sound (though I haven't checked proofs in Appendix carefully) and the experiments show improved performance over existing approaches. The authors do not claim anything that is not backed by evidence in the form of experiments and theory.

**Essential References Not Discussed:**

None that I am aware of.

**Experimental Designs Or Analyses:**

The experiments appear sound and there is no major issue with the evaluations.

**Methods And Evaluation Criteria:**

Yes. The authors use standard evaluation methods and use standard RL environments in the experiments.

**Other Comments Or Suggestions:**

- I think the term "secret MDP" is a bit confusing as it suggests that there is a secret environment in which we want the agent to perform well. Something like task-selection MDP or meta-MDP is maybe less confusing.
- As someone with no prior background on MPTS, it was a little hard to figure out which parts are "new" in this paper. I had to look at the MPTS paper to clearly figure this out. It would be great if this can be clarified in the paper.

**Other Strengths And Weaknesses:**

### Additional Strengths

- The experimental results appear convincing. The authors show clear improvements in both (i) the final performance and (ii) the sample-efficiency of learning.
- The approach is shown to be lightweight and offers performance improvements with minimal impact to cost.
- The paper is well-written and prior concepts related to MPTS are explained well.

There does not appear to be any major weakness that would impact acceptance. Some minor comments are below.

**Questions For Authors:**

N/A

**Relation To Broader Scientific Literature:**

- Overall, framing the task-selection problem as an RL problem appears novel and could potentially provide a framework for future work on meta learning. The authors also showed how prior work like MPTS fits into their framework.
- Insight on task diversity and how the selection criterion can lead to selecting tasks from a narrow range of values is interesting and applicable more broadly.
- The proposed approach is agnostic to the specific meta-RL algorithm used and the authors show this using experiments based on different meta-RL algorithms in the literature.

**Theoretical Claims:**

I did not reach the proofs in the Appendix (only briefly glanced over them).

---

> ### Author Rebuttal · Authors · 2025-03-29
>
> **_We sincerely appreciate Reviewer beLW's efforts and recognition of our work. Below, we improve the manuscript based on beLW's feedback._**
> ___
>
> **1. Terminology clarity about secret MDP**
>
> Thank you for your valuable feedback. We used the term "secret MDP" to highlight that we are the first to model robust active task sampling as an MDP and solve it with developed i-MABs. We will further explain this point in the revised manuscript and consider using a more intuitive term, such as "task-selection MDP" to avoid potential confusion.
>
> **2. Comparison betwen PDTS and MPTS**
>
> Thank you for the suggestion. Currently, we primarily summarize the orthogonal contributions of MPTS and PDTS in Table 1. For example, MPTS develops a VAE-like risk predictive model to achieve nearly $\text{CVaR}_\alpha$ optimization but suffers from the concentration issue. Our PDTS (i) introduces *a versatile theoretical tool i-MAB* to achieve a more robust solution, i.e., nearly worst-case optimization for meta RL and DR cases, (ii) *resolve the concentration issue*, and (iii) offer *easier implementation with stochastic optimism*. These three contribution points are new and increase scalability of RATS. Encouragingly, PDTS with i-MABs provides stable and tractable scheme for worst-case optimization in adaptive decision-making, broadening its applicability.
>
> In the revised manuscript, we will involve more RATS background and further highlight the significance of PDTS for more general readers.
>
> ___
>
> **_Once again, thank you for your valuable review and thoughtful recognition of our work. Your feedback is greatly appreciated and has significantly improved our manuscript. We hope our responses sufficiently answer your questions._**

---

> > ### Comment · Reviewer_beLW · 2025-04-02
> >
> > I thank the authors for their detailed response which answered my questions. As mentioned in the other reviews, I would encourage the authors to also include a discussion on limitations in the paper.

---

> > > ### Author Response · Authors · 2025-04-02
> > >
> > > Sure ^.^. **We will take all suggestions into the revised version, including the mentioned empirical findings and limitation summary.** We hope the i-MABs in this work will facilitate the algorithm design of efficient robust adaptation and release the power of reinforcement learning in large-scale decision-making.
> > >
> > > Importantly, all suggestions and questions are constructive in improving our manuscript and we thank reviewers and area chairs efforts in this work.

---

### Official Review · Reviewer_v1xi · 2025-03-12

**Overall Recommendation:** 3

**Summary:**

The paper focuses on adaptation robustness, addressing scenarios where a risk-predictive model is utilized to mitigate intense evaluation requirements. It formulates the robust active task sampling (RATS) problem as a partially observable Markov decision process (POMDP), providing theoretical insights into the problem. Empirically, the paper demonstrates that the proposed method, Posterior-Diversity Synergized Task Sampling, achieves stronger performance in vision-based reinforcement learning tasks compared to baseline methods.

**Claims And Evidence:**

The claims are supported by empirical results.

One question I have is regarding the introduction, where the authors state that their method requires less complex configurations compared to prior works. Could the authors clarify what is meant by ‘configurations’? Providing additional context and specific examples would help in understanding this claim

**Essential References Not Discussed:**

N/A

**Experimental Designs Or Analyses:**

The experimental design appears sound to me.

A general question I have is regarding the limitations of the proposed method. What are the potential failure cases if the assumptions made in the analysis do not hold? A discussion on these aspects would help in understanding the robustness and applicability of the approach.

**Methods And Evaluation Criteria:**

The methods and evaluation protocol are mainly based on meta-RL literature and robust RL literature, which makes sense to me.

**Other Comments Or Suggestions:**

N/A

**Other Strengths And Weaknesses:**

N/A

**Questions For Authors:**

Please see previous sections.

**Relation To Broader Scientific Literature:**

Robust adaption is very important in real-world applications, especially robotics applications.

**Theoretical Claims:**

I did not evaluate the theoretical analysis as I am not familiar with this field.

---

> ### Author Rebuttal · Authors · 2025-03-29
>
> _**We sincerely appreciate Reviewer v1xi's efforts and positive feedback. Below, we provide our responses.**_
> ___
>
> **1. Clarification on the simplification of configurations**
>
> Apologies for any confusion. The simplification of configurations lies in two aspects:
> - As stated in Lines 261–270 of Section 3.2, MPTS requires careful tuning of the candidate task batch size $\hat{\mathcal{B}}$. In contrast, PDTS eliminates this requirement and remains scalable even under extreme worst-case optimization (e.g.,$\hat{\mathcal{B}} = 64 \times \mathcal{B}$) by introducing a diversity-regularized acquisition function to mitigate the concentration issue.
> - As stated in Lines 281–284 of Section 3.3, we use posterior sampling to utilize stochastic optimism while avoiding the calibration of exploration and exploitation weights in subset search, which is required by UCB-based methods.
>
> We will highlight these in the revised manuscript.
>
> **2. Discussion on assumptions failure**
>
> Thank you for your thoughtful question. The assumptions in our analysis are consistent with those in prior works[1,2], as stated in Appendix A. If they do not hold, the effectiveness of the risk predictive model may degrade with extremely lower Pearson correlation coefficients, potentially impacting robust optimization. Fortunately, our experiments demonstrate that PDTS achieves higher PCC values, strong robust optimization performance, empirically supporting the validity of these assumptions. We will incorporate this discussion into the revised manuscript for clarity.
>
> ___
>
> **_Once again, thank you for your valuable review. Your feedback is greatly appreciated and has helped improve our manuscript. We hope our responses adequately address your concerns, and we would be grateful if you could consider raising the score._**
>
> ___
> **References:**\
> [1] Greenberg I, Mannor S, Chechik G, et al. Train hard, fight easy: Robust meta reinforcement learning[J]. Advances in Neural Information Processing Systems, 2023, 36: 68276-68299.\
> [2] Wang, Q., Lv, Y., Xie, Z., & Huang, J. (2023). A simple yet effective strategy to robustify the meta learning paradigm. Advances in Neural Information Processing Systems, 36, 12897-12928.

---

### Official Review · Reviewer_b9NR · 2025-03-12

**Overall Recommendation:** 4

**Summary:**

This paper tackles robust active task sampling (RATS) in domain randomization or meta-RL for worst-case performance. Tasks are viewed as arms in an infinite multi-armed bandit, but the existing MPTS can over-concentrate on top-B tasks. The authors propose PDTS, which replaces UCB-based acquisition with posterior sampling and adds a diversity term. Experiments show PDTS achieves faster, more robust adaptation than baselines (ERM, DRM, GDRM, MPTS) on MuJoCo and domain-randomized robotics tasks.

**Claims And Evidence:**

Overall, the claims have coherent theoretical backing and empirically strong results across multiple benchmarks.

**Essential References Not Discussed:**

No major omissions jump out. The paper cites standard domain-randomization, risk-averse RL, and meta-learning literature.

**Experimental Designs Or Analyses:**

The experimental design is thorough and appropriate for the proposed method, supporting the authors’ conclusions about robust adaptation performance and sample efficiency.

**Methods And Evaluation Criteria:**

The evaluation criteria make sense for risk-averse policy adaptation. The chosen suite of MuJoCo and robotic tasks is widely used in domain randomization and meta-RL research, so the evaluation is aligned with standard practice.

**Other Comments Or Suggestions:**

- Clarifying the best practices for choosing the diversity weight $\gamma$ or the approximate search method might help practitioners.

- Additional ablations on how the quality of the risk-predictive model $p(\ell|\tau)$ influences PDTS’s final performance could further strengthen the discussion.

**Other Strengths And Weaknesses:**

- Strengths:

1. The i-MAB perspective is novel and successfully integrates RATS with risk-averse RL under a cohesive theoretical argument.

2. The PDTS method is simple but effectively addresses subset concentration by blending posterior sampling and diversity.

3. The empirical evaluation is robust, spanning multiple benchmarks (both symbolic and visual), and consistently demonstrates performance improvements.

- Weaknesses:

1. Relying on a large pseudo batch size $\hat{B}$ for nearly worst-case coverage can introduce computational overhead.

2. The new diversity regularization parameter $\gamma$ may require careful tuning.

3. The method heavily depends on a risk-predictive model. If that model’s performance degrades, PDTS coverage might fail to accurately capture the most challenging tasks. While correlation results are encouraging, model reliability remains a potential concern.

**Questions For Authors:**

1. Scaling to high dimensions: How does PDTS handle extremely high-dimensional task spaces? Do you have any heuristic or projection strategy if the dimension is large? This would clarify how widely PDTS can be applied in large-scale real-world DR or meta-RL scenarios.

2. Diversity Regularization: In practice, how sensitive is the method to the diversity weight $\gamma$? Would you expect the best $\gamma$ to scale with $\hat{B}$ or with dimension $d$? If so, how?

**Relation To Broader Scientific Literature:**

The paper situates itself among:

- Risk-averse RL frameworks (DRM/CVaR, GDRM, etc.).

- Meta-RL approaches.

**Theoretical Claims:**

All theoretical results are plausible and consistent with standard concepts in bandit theory and set diversification. There is no obvious flaw in these short formal statements. Given the scope of the paper, the claims appear correct, and the sketches/logic are standard enough that no glaring issues stand out.

---

> ### Author Rebuttal · Authors · 2025-03-29
>
> _**We sincerely appreciate Reviewer b9NR's efforts and constructive feedback. Below, we provide our responses.**_
> ___
>
> **1. Additional computational overhead**
>
> Thanks for precious comment. We quantitatively analyzed the extra computational overhead in Fig.5. Even with a $64\times$ candidate batch (i.e., $\text{CVaR}_{1-1/64}$ approximating the worst case), **the additional computational overhead remains negligible due to the efficiency of the risk predictive model—its cost is significantly lower than that of agent-environment interactions and policy optimization in MetaRL or DR**. We will emphasize this point in the revised manuscript.
>
> **2. Ablation study on the risk predictive model**
>
> Thank you for the valuable question—it has been very helpful to our analysis. We conducted an ablation study to analyze the impact of the risk predictive model. We designed two variants, **PDTS-Deep and PDTS-Shallow**, by increasing and decreasing the number of encoder-decoder layers, respectively. Additionally, we replaced the encoder-decoder structure with an MLP to create **PDTS-MLP**. Results on Walker2dVel are summarized in the table below:
>
> |Methods|$\text{CVaR}_{0.9}$|$\text{CVaR}_{0.7}$|$\text{CVaR}_{0.5}$|$\text{Average}$|
> |-|-|-|-|-|
> |ERM|-69.77$\pm$7.62|-31.73$\pm$7.82|-3.78$\pm$6.66|38.88$\pm$4.73|
> |PDTS|**-22.42$\pm$3.13**|**2.86$\pm$3.04**|16.57$\pm$2.93|40.40$\pm$3.07|
> |PDTS-Deep|-30.51$\pm$7.11|0.55$\pm$5.69|**17.99$\pm$4.8**|**44.42$\pm$3.69**|
> |PDTS-Shallow|-41.24$\pm$4.74|-11.34$\pm$4.5|3.92$\pm$4.33|33.64$\pm$4.01|
> |PDTS-MLP|-41.07$\pm$4.88|-12.04$\pm$5.06|3.38$\pm$4.94|32.96$\pm$4.58|
>
> From these results, we observe:
> - All PDTS variants achieve better task robust adaptation, confirming the effectiveness of PDTS and its generality across different risk predictive models.
> - The encoder-decoder architecture generally outperforms MLP-based models, supporting the rationale behind this design.
> - Deeper networks may introduce a performance-robustness trade-off in the current setting, which we plan to further investigate in more complex scenarios.
> - Weaker risk prediction models degrade overall performance, due to poorer difficult MDP identification.
>
> **3. Scaling to high dimensions**
>
> Thank you! This is a very insightful and important question. Since RATS is still in its early stages, we are actively exploring its scalability to high-dimensional task spaces. We agree that heuristic or projection strategies could be viable solutions. One potential approach is to leverage lightweight general-purpose embedding models, such as WordLLaMA[1], to compress high-dimensional task identifiers from language or vision modalities[2]. We appreciate your insight and will continue to investigate this direction further to broaden the application scope of PDTS.
>
> **4. Sensitivity of hyperparameter $\gamma$ and tuning practices**
>
> We conducted an ablation study on $\gamma$ in Figure 13 and found that **PDTS is relatively stable to $\gamma$ within a certain range**. Below, we summarize two **key tuning recommendations**
> - In most cases, setting $\gamma$ to 1 or a nearby value secures superior enough performance.
> - For scenarios with an extremely low-dimensional task identifier, increasing $\gamma$ appropriately may improve performance.
>
> We will incorporate these recommendations into the revised manuscript and specify the $\gamma$ values used in practice in the released code to aid practitioners.
>
> **5. Would you expect the best $\gamma$ to scale with $\hat{\mathcal{B}}$ or with dimension $d$?**
>
> We suggest using a larger candidate batch, $\hat{\mathcal{B}}$, to better capture the worst-case scenario and improve performance, as shown in Figure 13. Therefore, we set $\hat{\mathcal{B}} = 64 \times \mathcal{B}$ in all main experiments without delicate tuning and believe it's unnecessary to co-tune it with $\gamma$. We hypothesize that **$\gamma$ scales positively with $\hat{\mathcal{B}}$**, as larger batches exacerbate concentration issues.
>
> Additionally, we expect that, in comparable scenarios, **a smaller $d$ leads to a larger $\gamma$**, as concentration issues become more likely. For example, we use $\gamma = 1$ for the task with 2D task identifier (Walker2dMassVel) and $\gamma = 5$ for the 1D task (Walker2dVel).
>
>
> ___
>
> _**Once again, thank you for your valuable review and thoughtful recognition of our work. Your feedback is greatly appreciated and has significantly improved our manuscript. We hope our responses sufficiently address your concerns.**_
>
> ___
>
> **References:**\
> [1] Miller, D. L. (2024). WordLlama: Recycled token embeddings from large language models.  https://github.com/dleemiller/wordllama \
> [2] Kim M J, Pertsch K, Karamcheti S, et al. Openvla: An open-source vision-language-action model[J]. arXiv preprint arXiv:2406.09246, 2024.

---

> > ### Comment · Reviewer_b9NR · 2025-04-09
> >
> > I thank the authors for their detailed response, which satisfactorily addressed my questions. I am maintaining my initial score for acceptance.

---

> > > ### Author Response · Authors · 2025-04-09
> > >
> > > We thank Reviewer b9NR for constructive suggestions and kind replies once again. We'll incorporate the mentioned discussions and statistical results in the updated manuscript.

---

### Official Review · Reviewer_xw4Y · 2025-03-13

**Overall Recommendation:** 4

**Summary:**

This paper studies a robust active task sampling (RATS) paradigm, models it as an infinitely many-armed bandit (i-MAB) problem, and proposes a novel method called Posterior and Diversity Synergized Task Sampling (PDTS). PDTS mitigates the task concentration issues in an existing approach, Model Predictive Task Sampling (MPTS), by incorporating a diversity regularized acquisition function, replacing the upper confidence bound acquisition function in MPTS. As a result, PDTS enables exploration in a broader range of tasks and improves robustness for nearly worst-cases. The authors conduct extensive experiments in various meta RL settings, and show that PDTS improves CVaR robustness, sample-efficiency for average return, and zero-shot adaptation in out-of-distribution tasks.

**Claims And Evidence:**

Claims in terms of mitigating the task concentration issue in MPTS, computational efficiency, versatility of the framework, improved robustness, and zero-shot adaptation are well supported through theoretical and empirical evidence. Although theoretically justified, empirical evidence supporting proposition 3.4 is missing. An interesting ablation would be observing the trade-off between improved nearly worst-case robustness and computational efficiency of PDTS.

**Essential References Not Discussed:**

The paper discusses various task sampling methods for providing risk-averseness and robustness, yet it does not mention curriculum learning at all. For example, in the paper that presents RoML (Greenberg et. al.), which the authors evaluate in their documents, curriculum learning methods are also studied and discussed. I believe a discussion of curriculum learning methods would broaden the audience of this work. Here are a few of those instances:

Dennis, M., Jaques, N., Vinitsky, E., Bayen, A., Russell, S., Critch, A., & Levine, S. (2020). Emergent complexity and zero-shot transfer via unsupervised environment design. Advances in neural information processing systems, 33, 13049-13061.

Jiang, M., Dennis, M., Parker-Holder, J., Foerster, J., Grefenstette, E., & Rocktäschel, T. (2021). Replay-guided adversarial environment design. Advances in Neural Information Processing Systems, 34, 1884-1897.

Koprulu, C., Simão, T. D., Jansen, N., & Topcu, U. (2023, July). Risk-aware curriculum generation for heavy-tailed task distributions. In Uncertainty in Artificial Intelligence (pp. 1132-1142). PMLR.

**Experimental Designs Or Analyses:**

I checked the experimental designs behind emta RL, domain randomization, and visual domain randomization settings. I haven't seen any issues.

**Methods And Evaluation Criteria:**

They make sense when evaluating CVaR robustness, sample-efficiency, and zero-shot adaptation in meta RL.

**Other Comments Or Suggestions:**

I didn't see any typos. However, I highly recommend that the authors refine their use of symbols. Section 2, and most importantly, Section 3, is very hard to read.

**Other Strengths And Weaknesses:**

Strengths:
- The paper clearly structures and explains the motivation, the problem, the proposed method and the experimental results.
- Introduction of i-MABs as a model for RATS provides a versatile theoretical framework.
- PDTS mitigates key issues in an existing method, MPTS, and provides computational efficiency.
- Theoretical results are clearly explained, and empirical evaluation presents strong evidence in favor of the proposed method in improving risk-averseness, robustness and generalization capabilities of meta RL agents.

Weaknesses:

- The flow of Section 2 and 3 degrades as more notation is used.
- There is no ablation study to justify nearly worst-case optimization, as proposition 3.4 suggests. Although the authors prove the proposition, an interesting study would be on the trade-off between worst-case performance of PDTS and the computational costs of increasing the cardinality of the set of candidate tasks.
- There is no discussion on the limitations of the introduced model and the proposed method.

**Questions For Authors:**

Does the meta RL agent have access to task identifiers? I assume it does not. But then I'm confused how they can be utilized in Eq. (11) to measure the diversity of candidate tasks. As the task identifiers are assumed to be partially observable in meta RL, I think a clarification is needed here.

**Relation To Broader Scientific Literature:**

The i-MAB model provides a theoretical framework for active task sampling, which is useful for risk-averseness in many decision-making problems, such as reinforcement learning and robotics, where robust adaptation is key for real-world applicability.

**Theoretical Claims:**

Yes, I checked the correctness of proofs of propositions 3.2, 3.3, and 3.4. I haven't seen any issues.

---

> ### Author Rebuttal · Authors · 2025-03-29
>
> _**We sincerely thank Reviewer xw4Y's efforts and thoughtful feedback. Below, we provide our responses.**_
> ___
>
> **1. Discussion of curriculum learning methods**
>
> Thank you for your valuable suggestion. We'll cite these important works and discuss them in the revised manuscript, adding contents below to Line 667:
>
> > Curriculum learning is also a crucial topic related to adaptive decision-making. Dennis, Michael, et al. [1] develop unsupervised environment design (UED) as a novel paradigm for environment distribution generation and achieve SOTA zero-shot transfer. Jiang, Minqi, et al. [2] cast prioritized level replay to enhance UED and formulate dual curriculum design for improving OOD and zero-shot performance. In [3], heavy-tailed distributions are incorporated into the automated curriculum, which leads to robustness improvement. In contrast, our work emphasizes robust task adaptation. Integrating the idea of surrogate evaluation from PDTS into curriculum design could be an interesting direction for future research.
>
> **2. Writing optimization**
>
> Thank you for your constructive suggestion. We will further simplify the math notations in Sections 2 and 3 to improve comprehension. For instance, instead of introducing the new notation $\mathbf{R}$, we will use the more intuitive summation form of the cumulated return to enhance clarity.
>
> **3. Trade-off between worst-case performance and the computational costs**
>
> Thanks for the insightful comment. This is a crucial point that we also investigated in Lines 432–438, as well as Figures 5 and 13. Figure 5 demonstrates that even with a $64\times$ candidate batch (i.e., $\text{CVaR}_{1-1/64}$ approximating the worst case), the **additional computational overhead remains negligible due to the efficiency of the risk predictive model—its cost is significantly lower than that of agent-environment interactions and policy optimization in MetaRL or DR**. Figure 13 shows that increasing the candidate batch size improves nearly worst-case performance. Therefore, we adopt the $64\times$ pseudo batch consistent in all main experiments without extra adjustments. We will emphasize this point in the revised manuscript.
>
> **4. Limitations discussion**
>
> Thank you for your insightful feedback. We briefly discussed the limitations of the proposed method in the Conclusion (Line 435-438) and Appendix E (Line 1347-1348). Specifically, our approach (1) relies on the risk predictive model, for roughly scoring task difficulties, and (2) depends on identifier information and the inherent smoothness of the adaptation risk function, which might not hold in restricted scenarios. We will provide a more detailed discussion in the revised manuscript.
>
> **5. Meta-RL agent have access to task identifiers**
>
> Apologies for the confusion. We will clarify this in the revised manuscript: The task identifiers are indeed visible in Meta-RL, as is consistent with prior works such as ROML [4] and MPTS [5]. Detailed information on the task identifiers can be found in Table 3.
>
> ___
>
> _**Once again, thank you for your valuable review. Your feedback is greatly appreciated and has helped improve our manuscript a lot. We hope our responses adequately address your concerns, and we would be grateful if you would reconsider the evaluation and update the score accordingly.**_
>
> ___
>
> **References:**\
> [1] Dennis, M., Jaques, N., Vinitsky, E., Bayen, A., Russell, S., Critch, A., & Levine, S. (2020). Emergent complexity and zero-shot transfer via unsupervised environment design. Advances in neural information processing systems, 33, 13049-13061.\
> [2] Jiang, M., Dennis, M., Parker-Holder, J., Foerster, J., Grefenstette, E., & Rocktäschel, T. (2021). Replay-guided adversarial environment design. Advances in Neural Information Processing Systems, 34, 1884-1897.\
> [3] Koprulu, C., Simão, T. D., Jansen, N., & Topcu, U. (2023, July). Risk-aware curriculum generation for heavy-tailed task distributions. In Uncertainty in Artificial Intelligence (pp. 1132-1142). PMLR.\
> [4] Greenberg I, Mannor S, Chechik G, et al. Train hard, fight easy: Robust meta reinforcement learning[J]. Advances in Neural Information Processing Systems, 2023, 36: 68276-68299.\
> [5] Wang Q C, Xiao Z, Mao Y, et al. Beyond Any-Shot Adaptation: Predicting Optimization Outcome for Robustness Gains without Extra Pay[J]. arXiv preprint arXiv:2501.11039, 2025.

---

> > ### Comment · Reviewer_xw4Y · 2025-04-09
> >
> > Thank you for your response. My concerns and questions have been adequately addressed. I will change my score to Accept. I hope to see the changes promised by the authors in the final version, as I believe they will greatly improve the readers' experience.

---

> > > ### Author Response · Authors · 2025-04-09
> > >
> > > We thank Reviewer xw4Y for insightful comments. We'll polish the manuscript and incorporate these suggestions inton the updated version as mentioned.

---

### Decision · Program_Chairs · 2025-05-01

**Decision:**

Accept (poster)

**Comment:**

All reviewers agree that the paper is worth publication at ICML. For this reason, I recommend acceptance. The authors are encouraged to revise the manuscript by incorporating all reviewer suggestions, including empirical findings, limitations, and statistical results, while also polishing the writing for clarity.